# Refining Dual Spectral Sparsity in Transformed Tensor Singular Values

## Abstract

The Tensor Nuclear Norm (TNN), derived from the tensor Singular Value Decomposition, is a central low-rank modeling tool that enforces *element-wise sparsity* on frequency-domain singular values and has been widely used in multi-way data recovery for machine learning and computer vision. However, as a direct extension of the matrix nuclear norm, it inherits the assumption of *single-level spectral sparsity*, which strictly limits its ability to capture the *multi-level spectral structures* inherent in real-world data—particularly the coexistence of low-rankness within and sparsity across frequency components. To address this, we propose the tensor $\ell_p$-Schatten-$q$ quasi-norm ($p, q \in (0, 1]$), a new metric that enables *dual spectral sparsity control* by jointly regularizing both types of structure. While this formulation generalizes TNN and unifies existing methods such as the tensor Schatten-$p$ norm and tensor average rank, it differs fundamentally in modeling principle by coupling global frequency sparsity with local spectral low-rankness. This coupling introduces significant theoretical and algorithmic challenges. To tackle these challenges, we provide a theoretical characterization by establishing the first minimax error bounds under dual spectral sparsity, and an algorithmic solution by designing an efficient reweighted optimization scheme tailored to the resulting nonconvex structure. Numerical experiments demonstrate the effectiveness of our method in modeling complex multi-way data.

## 1 Introduction

Modeling latent structural patterns in high-dimensional signals is a fundamental challenge across domains such as machine learning and signal processing [17, 38, 19]. Real-world datasets are often inherently multi-modal and high-dimensional (tensor-form), containing intricate dependencies that cannot be adequately captured by naïve modeling or vector/matrix-based representations [4]. A common strategy to uncover these relationships is to impose a *low-rank* prior, which isolates essential information and reduces the degrees of freedom, focusing on the principal components of the signal [21, 1]. Traditional tensor decomposition methods, such as CANDECOMP/PARAFAC (CP) [3], Tucker [27], and Tensor Train [23], have been widely used to model tensor signals [4, 16, 8, 34]. While effective in certain scenarios, these methods rely on the assumption of intrinsic low-rankness in the *original domain*, which may fail to hold in complex, real-world applications. This limitation has led to the development of *transformed-domain* modeling, where linear transformations like the Discrete Fourier Transform (DFT) are applied to reveal more pronounced low-rank patterns. Within this paradigm, the tensor Singular Value Decomposition (t-SVD) has emerged as a powerful framework with notable success in applications such as image and video analysis [17, 38, 32, 30].

Building on the t-SVD framework, the Tensor Nuclear Norm (TNN) has become an extensively adopted regularizer for low-rank tensor modeling [20, 38, 25, 6, 36, 18, 39]. By extending the matrix

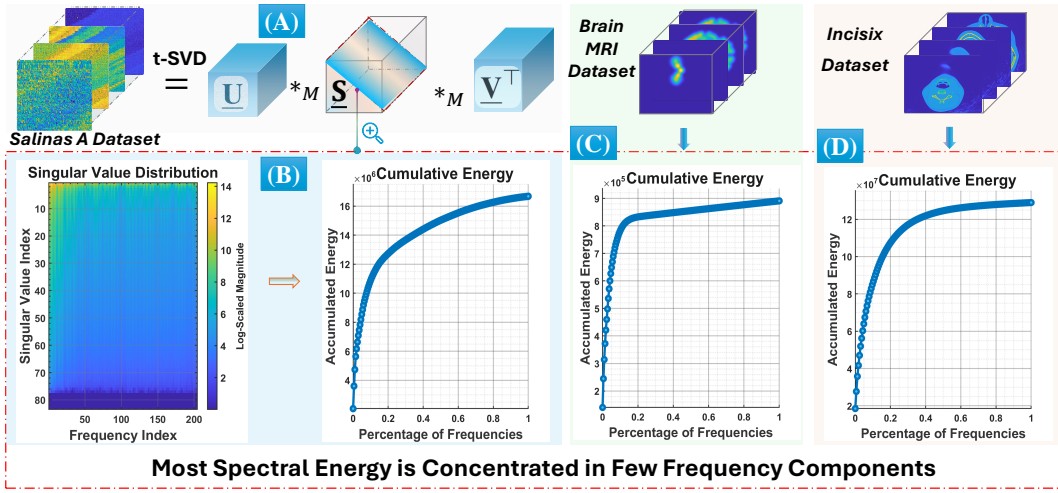

Figure 1: Empirical illustration of dual spectral sparsity patterns in the transformed (DCT) domain via t-SVD. (A) The t-SVD framework decomposes a tensor into frequency-domain singular structures. (B)-Left: Singular value heatmap of the *Salinas A* dataset under t-SVD—each column represents one frequency slice. Vertical decay reveals intra-frequency low-rankness, while horizontal variation indicates sparsity across frequencies. (B)-Right, (C), (D): Cumulative energy curves for *Salinas A*, *Brain MRI*, and *Incisix* datasets show that over 80% of total spectral energy is concentrated in the top 15%–30% frequency components, confirming frequency-wise sparsity. These observations support the presence of a dual-level spectral structure and motivate regularization schemes that go beyond uniform norms like TNN [38, 19] to jointly model frequency sparsity and low-rankness.

nuclear norm to the tensor setting, TNN promotes low-rankness by enforcing *element-wise sparsity* on singular values in the transformed domain [13, 38]. This formulation effectively captures low-rank dependencies within individual frequency components.

However, a long-overlooked limitation of TNN lies in its assumption of *uniform spectral regularization*, which treats all frequency components equally regardless of their relative importance. From a signal processing perspective, this *single-level sparsity* design fails to account for the dual-level structure often observed in transformed tensor data. In particular, real-world tensors may exhibit strong low-rankness within each frequency component along with sparsity across the frequency domain. As illustrated in Fig.1 and further discussed in §3, empirical analyses of several datasets, including hyperspectral images and medical imaging volumes, indicate that a small subset of frequency slices contributes the majority of spectral energy. In addition, these dominant components often exhibit pronounced low-rank structures within each frequency slice. These observations suggest the need for a more flexible regularization framework that can separately characterize both intra-frequency low-rankness and inter-frequency sparsity, instead of relying on a uniform scheme like TNN.

These limitations necessitate a new method capable of modeling both levels of sparsity. This raises three interconnected questions:

**RQ1 (Modeling):** *how to effectively model both intra-frequency and inter-frequency dependencies in tensor data?*

**RQ2 (Theory):** *can we establish rigorous theoretical guarantees to validate such a framework, given the challenges of analyzing coupled sparsity?*

**RQ3 (Algorithm):** *can efficient algorithms be designed to tackle the optimization challenges introduced by the coupled sparsity structure?*

To address these questions, we propose the *tensor $\ell_p$-Schatten-q quasi-norm*, a novel framework introducing *dual spectral sparsity control* to simultaneously model both within-frequency and across-frequency dependencies. Specifically, parameter $p$ governs sparsity among different frequency components (**RQ1**), while parameter $q$ controls low-rankness within each frequency component. This framework generalizes and extends TNN, unifying existing methods such as the tensor Schatten-$p$ quasi-norm [12] and tensor average rank [31] into a single, versatile framework.

While our framework offers promising modeling capabilities, the *coupled nature of this dual spectral sparsity* introduces significant theoretical and computational challenges. Our main contributions in developing and validating this framework are as follows:

- **Structural Modeling (RQ1):** To the best of our knowledge, this work is the first to rigorously formalize and explicitly model a coupled spectral structure within the t-SVD framework, where inter-frequency sparsity coexists with intra-frequency low-rankness (Section 3). The proposed $\ell_p$-Schatten-$q$ quasi-norm jointly models both inter-frequency sparsity and intra-frequency low-rankness, while allowing separate control over each via parameters $p$ and $q$. This formulation captures hierarchical spectral structure beyond uniform regularizers such as TNN.

- **Theoretical Guarantees (RQ2):** We establish sharp minimax lower and upper bounds for tensor estimation under dual spectral sparsity, covering both hard and soft regimes (Section 4). The analysis introduces new techniques to characterize the complexity of coupled parameter spaces, extending classical tools such as covering numbers and metric entropy to the tensor spectral setting.

- **Optimization and Empirical Validation (RQ3):** We develop a scalable proximal algorithm tailored to the proposed quasi-norm (Section 5). It employs a reweighted $\ell_{1/2}$ approximation and frequency-wise singular value updates in the transform domain, effectively handling the nonconvexity and structural coupling induced by dual spectral sparsity. Experiments on real-world tensor recovery tasks demonstrate the potential applicability of our method (Section 6).

The remainder of the paper is organized as follows. Section 2 reviews basic preliminaries. Section 3 introduces the proposed quasi-norm. Sections 4 and 5 present the theoretical analysis and optimization algorithm, respectively. Experimental results are reported in Section 6, followed by the conclusion in Section 7. Details on related work, proofs, algorithms, and experiments are provided in the appendix.

## 2 Notations and Preliminaries

**Notations.** For any positive integer $d$, let $[d] := \{1, \ldots, d\}$. We denote vectors by lowercase bold letters (e.g., $\mathbf{a}$), matrices by uppercase bold letters (e.g., $\mathbf{A}$), and 3-way tensors by underlined uppercase letters (e.g., $\underline{\mathbf{A}}$). Constants, represented as $c$ and its variants (e.g., $c_1$, $C$), may vary in value across contexts. For a 3-way tensor of size $d_1 \times d_2 \times m$, we assume $d_1 \geq d_2$ without loss of generality.

For a matrix $\mathbf{A} \in \mathbb{R}^{d_1 \times d_2}$, we define $\boldsymbol{\sigma}(\mathbf{A})$ as the vector of its singular values, arranged in descending order. The spectral norm $\|\mathbf{A}\|_{\text{spec}}$ and nuclear norm $\|\mathbf{A}\|_*$ of $\mathbf{A}$ are defined as the largest and the sum of its singular values, respectively. For any tensor $\underline{\mathbf{A}}$, we define its $\ell_p$-norm as $\|\underline{\mathbf{A}}\|_p := \|\operatorname{vec}(\underline{\mathbf{A}})\|_p$ and its Frobenius norm as $\|\underline{\mathbf{A}}\|_{\text{F}} := \|\operatorname{vec}(\underline{\mathbf{A}})\|_2$, where $\operatorname{vec}(\cdot)$ denotes the vectorization operation [11]. The inner product of two tensors $\underline{\mathbf{A}}$ and $\underline{\mathbf{B}}$ is given by $\langle \underline{\mathbf{A}}, \underline{\mathbf{B}} \rangle := \operatorname{vec}(\underline{\mathbf{A}})^\top \operatorname{vec}(\underline{\mathbf{B}})$. For a tensor $\underline{\mathbf{A}} \in \mathbb{R}^{d_1 \times d_2 \times m}$, we denote its $i$-th frontal slice as $\underline{\mathbf{A}}_{:,:,i}$ or simply $\underline{\mathbf{A}}_i$ when clear from context.

**The t-SVD Framework.** The t-SVD framework is based on the t-product operation, a generalization of matrix multiplication to tensors, which operates under an invertible linear transform $M$ [9]. By enhancing low-rank properties through specific linear transformations, this approach effectively exploits intrinsic correlations within the data [36, 29]. This paper adopts the convention of using orthogonal matrices for $M$ due to their stability and computational advantages [18, 28]. Specifically, for an orthogonal matrix $\mathbf{M} \in \mathbb{R}^{m \times m}$, we define the $M$-linear transform and its inverse on a tensor $\underline{\mathbf{T}} \in \mathbb{R}^{d_1 \times d_2 \times m}$ as:

$$M(\underline{\mathbf{T}}) := \underline{\mathbf{T}} \times_3 \mathbf{M}, \quad \text{and} \quad M^{-1}(\underline{\mathbf{T}}) := \underline{\mathbf{T}} \times_3 \mathbf{M}^{-1}, \tag{1}$$

where $\times_3$ denotes the mode-3 tensor-matrix product [9]. Using this transform, we introduce the basic notions in the t-SVD framework.

**Definition 2.1** (t-product [9]). The t-product of two tensors $\underline{\mathbf{A}} \in \mathbb{R}^{d_1 \times d_2 \times m}$ and $\underline{\mathbf{B}} \in \mathbb{R}^{d_2 \times d_3 \times m}$ under the transform $M$ in (1) is denoted by $\underline{\mathbf{A}} *_M \underline{\mathbf{B}} = \underline{\mathbf{C}} \in \mathbb{R}^{d_1 \times d_3 \times m}$, where $M(\underline{\mathbf{C}}) = M(\underline{\mathbf{A}}) \odot M(\underline{\mathbf{B}})$ in the transformed domain, and $\odot$ denotes the frontal-slice-wise product of the tensors.

**Definition 2.2** ($M$-block-diagonal matrix [28]). For a tensor $\underline{\mathbf{T}} \in \mathbb{R}^{d_1 \times d_2 \times m}$, its $M$-block-diagonal matrix $\bar{\mathbf{T}} \in \mathbb{R}^{d_1 m \times d_2 m}$ is defined as

$$\bar{\mathbf{T}} := \operatorname{bdiag}(M(\underline{\mathbf{T}})) = \operatorname{diag}\left(M(\underline{\mathbf{T}})_{:,:,1}, \ldots, M(\underline{\mathbf{T}})_{:,:,m}\right),$$

where $M(\underline{\mathbf{T}})$ is the mode-3 transform of $\underline{\mathbf{T}}$, and the operator $\operatorname{bdiag}(\cdot)$ stacks the frontal slices as diagonal blocks.

115  We now formally introduce the t-SVD, as illustrated in Fig. 1-(A).

116  **Definition 2.3** (t-SVD and tensor tubal rank [9])**.** The tensor Singular Value Decomposition (t-SVD)
117  of a tensor $\underline{\mathbf{T}} \in \mathbb{R}^{d_1 \times d_2 \times m}$ under the invertible linear transform $M$ in (1) is:

$$\underline{\mathbf{T}} = \underline{\mathbf{U}} *_M \underline{\mathbf{S}} *_M \underline{\mathbf{V}}^{\top}, \tag{2}$$

118  where $\underline{\mathbf{U}} \in \mathbb{R}^{d_1 \times d_1 \times m}$ and $\underline{\mathbf{V}} \in \mathbb{R}^{d_2 \times d_2 \times m}$ are t-orthogonal tensors, and $\underline{\mathbf{S}} \in \mathbb{R}^{d_1 \times d_2 \times m}$ is an
119  f-diagonal tensor. The tubal rank of $\underline{\mathbf{T}}$ is defined as the number of non-zero tubes in $\underline{\mathbf{S}}$ in the t-SVD,
120  i.e., $r_{\text{tb}}(\underline{\mathbf{T}}) := \#\{i \mid \underline{\mathbf{S}}_{i,i,:} \neq \mathbf{0}, i \leq \min\{d_1, d_2\}\}$.

121  To further model the low-rank structure of tensors in the transformed domain, the tensor nuclear norm
122  (TNN) is proposed as a key regularizer in low-rank tensor learning:

123  **Definition 2.4** (Tensor nuclear norm [20])**.** The tensor nuclear norm (TNN) of a tensor $\underline{\mathbf{T}} \in$
124  $\mathbb{R}^{d_1 \times d_2 \times m}$ under the transform $M$ are defined as $\|\underline{\mathbf{T}}\|_* := \|\bar{\mathbf{T}}\|_* = \|\boldsymbol{\sigma}(\bar{\mathbf{T}})\|_1$.

125  In this definition, TNN captures *the element-wise sparsity of the transformed spectrum* $\boldsymbol{\sigma}(\bar{\mathbf{T}}) \in$
126  $\mathbb{R}^{m \cdot \min\{d_1, d_2\}}$, allowing it to promote low-rank characteristics in the spectral domain. This property
127  has made TNN a foundational tool in tensor analysis, particularly for low-rank tensor recovery in
128  various applications such as image inpainting [19].

# 3 Dual Spectral Sparsity in the t-SVD Framework

130  Effectively capturing both intra-frequency low-rankness and inter-frequency sparsity (**RQ1**) is es-
131  sential for modeling structured tensor data. While methods like TNN emphasize within-frequency
132  low-rankness, they overlook sparsity across frequencies, limiting their ability to represent hierar-
133  chical dependencies. To overcome this, we introduce the $\ell_p$-Schatten-$q$ quasi-norm, a dual-sparsity
134  regularization framework designed to capture both levels of structure in a unified way.

135  **Limitations of TNN from a Group Sparsity Perspective.** According to Definition 2.4, the tensor
136  nuclear norm (TNN) promotes low-rankness by enforcing element-wise sparsity on singular values in
137  the transformed domain, effectively capturing intra-frequency low-rank structures. However, it applies
138  uniform regularization across all frequency components, regardless of their spectral importance. This
139  design fails to exploit the potential sparsity across frequency slices that is often present in real-world
140  tensors. Fig. 1 presents empirical evidence from three representative datasets—*Salinas A*[1], *Brain MRI*
141  [33], and *Incisix* [5]—demonstrating that only a small portion of frequency components accounts
142  for the majority of spectral energy. Specifically, more than 80% of the energy is concentrated in the
143  top 15%–30% of frequency bands. Meanwhile, the singular value heatmap (Fig. 1(B)-Left) reveals
144  pronounced horizontal sparsity, indicating that many frequency slices contribute minimally. Within
145  each active frequency slice, singular values decay rapidly, confirming low-rankness.

146  These observations suggest a dual-level structure comprising inter-frequency sparsity and intra-
147  frequency low-rankness. From a group sparsity perspective, the spectrum $\boldsymbol{\sigma}(\bar{\mathbf{T}})$ can be partitioned
148  into groups, where each group corresponds to the singular values $\boldsymbol{\sigma}(M(\underline{\mathbf{T}})_{:,:,i})$ of a specific frequency
149  slice. TNN enforces uniform regularization across these groups, overlooking their heterogeneous
150  importance. As a result, it may underperform when modeling data with hierarchical spectral structures.
151  These limitations motivate a more expressive framework that separately accounts for both levels of
152  structure.

153  **Hard Dual Spectral Sparsity.** To address the limitations of TNN, we first define a hard dual spectral
154  sparsity structure, where the tensor is assumed to satisfy exact sparsity constraints across and within
155  frequency components. This serves as an idealized formulation that captures the extreme case of dual
156  spectral sparsity and provides a clean theoretical foundation for later analysis.

157  **Definition 3.1** (Hard Dual Spectral Sparsity)**.** A tensor $\underline{\mathbf{T}} \in \mathbb{R}^{d_1 \times d_2 \times m}$ is said to exhibit $(s, r)$-dual
158  sparsity under a linear transform $M$ if it satisfies two constraints:

159  ***I.*** *Inter-frequency sparsity:* The number of active frequency components is limited to at most $s$.
160  Specifically, only $s$ out of the $m$ frequency components can have non-zero singular value vectors:
161  $\sum_{i=1}^{m} \mathbb{I}(\boldsymbol{\sigma}(M(\underline{\mathbf{T}})_{:,:,i}) \neq \mathbf{0}) \leq s$, where $\boldsymbol{\sigma}(M(\underline{\mathbf{T}})_{:,:,i})$ denotes the singular value vector of the $i$-th
162  frontal slice in the transformed domain.

---

[1] https://www.ehu.eus/ccwintco/index.php?title=Hyperspectral_Remote_Sensing_Scenes

163  ***II.*** *Intra-frequency low-rankness:* Within each active frequency component, the number of non-zero
164  singular values is constrained to at most $r$. This condition ensures a low-rank structure for
165  each frequency slice ($\forall i \in [m]$): $\sum_{j=1}^{\min\{d_1,d_2\}} \mathbb{I}\left(\sigma_j(M(\underline{\mathbf{T}})_{:,:,i}) \neq 0\right) \leq r$, where $\sigma_j(M(\underline{\mathbf{T}})_{:,:,i})$
166  denotes the $j$-th singular value of the $i$-th frontal slice of $M(\underline{\mathbf{T}})$.

167  This definition captures a strict form of dual-level structure by simultaneously enforcing sparsity
168  across frequencies and low-rankness within each active frequency slice. While such hard constraints
169  may be too restrictive in practical scenarios, especially where spectral contributions decay grad-
170  ually, they provide a clear conceptual framework to motivate and analyze the more flexible soft
171  regularization.

172  **Soft Dual Spectral Sparsity.** While the hard dual spectral sparsity model provides a clean conceptual
173  foundation, its strict assumption of exact sparsity and fixed-rank constraints is often impractical in
174  real-world scenarios. In many cases, singular values decay gradually rather than drop abruptly to
175  zero, and the true number of active frequency components may be ambiguous or noise-sensitive. To
176  overcome these limitations, we introduce a soft relaxation that allows for approximate sparsity and
177  low-rankness in a continuous manner. Specifically, we propose the $\ell_p$-Schatten-$q$ quasi-norm, which
178  relaxes the hard dual-sparsity constraints into a soft dual spectral sparsity framework.

179  **Definition 3.2** (Tensor $\ell_p$-Schatten-$q$ quasi-norm). For a tensor $\underline{\mathbf{T}} \in \mathbb{R}^{d_1 \times d_2 \times m}$, we define its tensor
180  $\ell_p$-Schatten-$q$ quasi-norm (abbreviated as $\ell_p(S_q)$-norm) as:

$$\|\underline{\mathbf{T}}\|_{\ell_p(S_q)} := \left( \sum_{i=1}^{m} \left( \sum_{j=1}^{d_1 \wedge d_2} \sigma_j(M(\underline{\mathbf{T}})_{:,:,i})^q \right)^{\frac{p}{q}} \right)^{\frac{1}{p}}, \tag{3}$$

181  where the exponents $(p,q) \in (0,1]^2$.

182  In this quasi-norm, $p$ governs the inter-frequency sparsity by promoting a group-wise regularization
183  across frequency components, effectively highlighting significant groups while suppressing others.
184  Simultaneously, $q$ controls the intra-frequency low-rankness by encouraging sparsity in the singular
185  values within each frequency slice, thereby modeling the intrinsic low-rank structure of the data.
186  This soft dual spectral sparsity framework provides a unified yet versatile approach to address the
187  hierarchical complexity of tensor data.

188  The $\ell_p$-Schatten-$q$ quasi-norm encompasses several existing regularization methods: it recovers TNN
189  when $(p,q) = (1,1)$[20], approximates the average rank as $(p,q) \to (1,0)$[31], and reduces to the
190  tensor Schatten-$q$ norm when $p = q$ [12], thereby offering greater modeling flexibility. *Despite*
191  *generalizing these regularizers, it fundamentally differs by jointly enforcing global frequency sparsity*
192  *and local spectral low-rankness.*

193  While TNN applies uniform regularization across all singular values, the $\ell_p$-Schatten-$q$ quasi-norm
194  introduces dual spectral sparsity control, modeling both inter-frequency sparsity through the $\ell_p$-quasi-
195  norm and intra-frequency low-rankness via the Schatten-$q$ quasi-norm. This dual-level flexibility
196  makes the proposed framework particularly well-suited for hierarchical and multi-scale data, where
197  dependencies and sparsity exhibit layered structures. By bridging the gap between element-wise
198  sparsity (as in TNN) and structured group sparsity, the $\ell_p$-Schatten-$q$ quasi-norm offers a more
199  expressive and adaptable approach, enabling precise control over structural patterns in modern
200  tensor-based analysis and recovery tasks.

201  # 4  Theory of Dual Spectral Sparse Tensor Estimation

202  This section develops the theoretical foundations of tensor estimation with dual spectral sparsity
203  structures (**RQ2**).

204  **Challenges.** Dual spectral sparsity, combining inter-frequency sparsity with intra-frequency low-
205  rankness, leads to *a globally coupled structure* that fundamentally differs from classical decoupled
206  models like TNN. The $\ell_p$-Schatten-$q$ quasi-norm imposes interdependent constraints across frequency
207  slices, resulting in a highly non-convex parameter space with nested sparsity patterns. This coupling
208  prohibits slice-wise decomposition and complicates the use of standard tools. Accurately characteriz-
209  ing the estimation complexity demands novel extensions of covering numbers and metric entropy
210  that jointly capture discrete sparsity and continuous low-rank structure.

To understand the statistical limits of learning under dual spectral sparsity, we analyze a simplified but representative model: the Gaussian location model, where the observed tensor is corrupted by additive noise. This setting preserves the core structural properties—inter-frequency sparsity and intra-frequency low-rankness—while avoiding complications unrelated to sparsity itself. Within this framework, we define structured parameter spaces that capture hard and soft variants of dual spectral sparsity, and establish sharp minimax lower and upper bounds under each. These results reveal how the joint effects of frequency selection and within-slice spectral decay determine the fundamental estimation limits, and provide theoretical justification for our proposed regularization.

## 4.1 Gaussian Location Model

Consider the Gaussian location model (GLM) [14], where $n$ independent noisy realizations of the target tensor $\underline{\mathbf{L}}^* \in \mathbb{R}^{d_1 \times d_2 \times m}$ are observed as:

$$\underline{\mathbf{Y}}_i = \underline{\mathbf{L}}^* + \underline{\mathbf{E}}_i, \quad i \in [n], \tag{4}$$

where $\underline{\mathbf{Y}}_i \in \mathbb{R}^{d_1 \times d_2 \times m}$ is the observed tensor, $\underline{\mathbf{L}}^*$ represents the ground truth tensor of interest, and $\underline{\mathbf{E}}_i \in \mathbb{R}^{d_1 \times d_2 \times m}$ denotes the noise tensor with entries independently drawn from $\mathcal{N}(0, \sigma^2)$. The parameter $\sigma$ characterizes the noise level. To simplify the analysis, we consider the sample mean of observations $\bar{\underline{\mathbf{Y}}} = n^{-1} \sum_{i=1}^n \underline{\mathbf{Y}}_i = \underline{\mathbf{L}}^* + \bar{\underline{\mathbf{E}}}$, where $\bar{\underline{\mathbf{E}}} = n^{-1} \sum_{i=1}^n \underline{\mathbf{E}}_i$ is the aggregated noise tensor with entries independently distributed as $\mathcal{N}(0, \sigma^2/n)$. The goal is to estimate the ground truth tensor $\underline{\mathbf{L}}^*$ based on the noisy observations $\{\underline{\mathbf{Y}}_i\}_{i=1}^n$. In particular, we aim to recover $\underline{\mathbf{L}}^*$ under dual spectral sparsity assumptions.

**Remark 4.1.** We adopt the Gaussian location model to isolate the core effects of dual spectral sparsity and the $\ell_p$-Schatten-$q$ regularization, avoiding additional complications from design tensors or sampling operators in tensor regression [35, 29, 24]. This simplified setting enables cleaner analysis and yields insights that extend naturally to regression problems under standard conditions such as RIP [35] or RSC [29, 24, 22].

**Dual Spectral Sparsity Assumptions.** We consider three distinct sparsity models for $\underline{\mathbf{L}}^*$:

*A1. Hard dual spectral sparsity*: Let $\underline{\mathbf{L}}^*$ belong to the parameter space

$$\mathsf{T}_{0,0}(s, r) = \{\underline{\mathbf{L}} : \text{at most } s \text{ active frequency slices, each of rank at most } r\}. \tag{5}$$

This model enforces exact inter-frequency sparsity and intra-frequency low-rankness.

*A2. Hard frequency sparsity and soft rank constraint (hard–soft sparsity)*: Let $\underline{\mathbf{L}}^*$ lie in

$$\mathsf{T}_{0,q}(s, R) = \left\{ \underline{\mathbf{L}} : |\{i : M(\underline{\mathbf{L}})_{:,:,i} \neq \mathbf{0}\}| \leq s, \ \|M(\underline{\mathbf{L}})_{:,:,i}\|_{S_q}^q \leq R, \ \forall i \in [m] \right\}. \tag{6}$$

This space imposes hard inter-frequency sparsity and soft Schatten-$q$ constraints within each active slice.

*A3. Soft dual spectral sparsity*: Let $\underline{\mathbf{L}}^*$ belong to the parameter space

$$\mathsf{T}_{p,q}(R) = \left\{ \underline{\mathbf{L}} : \|\underline{\mathbf{L}}\|_{\ell_p(S_q)}^p \leq R \right\}. \tag{7}$$

Here, $p$ promotes inter-frequency sparsity and $q$ controls intra-frequency low-rankness via spectral decay; $R$ specifies the quasi-norm ball radius.

These parameter spaces offer different views on structured tensor estimation: the *hard sparsity* model enforces strict thresholds, the *hard–soft model* balances structure with adaptability, and the *fully soft model* captures gradual spectral decay. Our goal is to estimate $\underline{\mathbf{L}}^*$ and derive minimax bounds under these assumptions.

## 4.2 Minimax Risk over Dual-level Sparse Structures

A key theoretical question in high-dimensional tensor estimation is: *What are the fundamental limits for recovering a tensor with dual spectral sparsity from noisy observations?* To address this, we establish minimax lower and upper bounds that characterize the best possible estimation accuracy achievable by any estimator under dual spectral sparsity assumptions.

$$\mathfrak{M}(\mathsf{T}) = \inf_{\hat{\underline{\mathbf{L}}}} \sup_{\underline{\mathbf{L}}^* \in \mathsf{T}} \mathbb{E}\left[ \|\hat{\underline{\mathbf{L}}} - \underline{\mathbf{L}}^*\|_{\mathrm{F}}^2 \right], \tag{8}$$

where $\mathsf{T}$ is the parameter space. Following [18, 19], we consider $d_1 = d_2 = d$ for simplicity.

**Theorem 4.2** (Minimax Bounds). *The minimax risk under dual spectral sparsity satisfies the following bounds under certain conditions[2]:*

**I.** *Hard constraints on both frequency sparsity and per-slice low-rankness:*

$$\mathfrak{M}(\mathbf{T}_{0,0}(s,r)) \asymp \frac{\sigma^2}{n}\left(s\log\frac{em}{s} + srd\right).$$

**II.** *Hard frequency sparsity with soft intra-slice Schatten-$q$ constraints:*

$$\mathfrak{M}(\mathbf{T}_{0,q}(s,R)) \asymp \frac{\sigma^2}{n}s\log\frac{em}{s} + sR\left(\frac{\sigma^2}{n}d\right)^{1-\frac{q}{2}}.$$

**III.** *Soft $\ell_p(S_q)$ constraints over both frequency and rank dimensions:*

$$\mathfrak{M}(\mathbf{T}_{p,q}(R)) \asymp \begin{cases} R\left(\frac{\sigma^2 n}{d}\right)^{\frac{p-2}{2}} + R\left(\frac{\sigma^2 n}{\log m}\right)^{\frac{p-2}{2}}, & p > q, \\ R^{\frac{q}{p}}\left(\frac{\sigma^2 n}{d}\right)^{\frac{q-2}{2}} + R\left(\frac{\sigma^2 n}{\log m}\right)^{\frac{p-2}{2}}, & p \le q,\ m > d^2, \\ R^{\frac{q}{p}}\left(\frac{\sigma^2 n}{d}\right)^{\frac{q-2}{2}}, & p \le q,\ m \le d^2. \end{cases}$$

Theorem 4.2 establishes the fundamental limits of estimation accuracy under different dual spectral sparsity structures. The minimax risk quantifies the worst-case squared Frobenius norm error that any estimator must incur when recovering a structured tensor from noisy observations. The results reveal the intricate balance between inter-frequency sparsity and intra-frequency low-rankness, showing how these factors jointly govern estimation complexity:

**I.** In the *hard sparsity* case, the estimation error consists of two terms: (i) $s\log(em/s)$, which reflects the difficulty of selecting $s$ active frequency components, and (ii) $srd$, which characterizes the challenge of estimating rank-$r$ matrices within each component.

**II.** In the *hard-soft sparsity* setting, the second term adapts to $sR(n^{-1}d)^{1-q/2}$, incorporating a smoother spectral decay controlled by $q$. Smaller $q$ values impose stronger low-rank constraints, effectively reducing estimation complexity by promoting more aggressive rank sparsity.

**III.** In the *fully soft sparsity* scenario, where both inter-frequency sparsity and intra-frequency rank constraints are relaxed, the minimax risk follows distinct scaling behaviors across regimes. When $p > q$, the error rate is dominated by $\ell_p$ sparsity, with $S_q$ low-rankness playing a minor role. For $p \le q$ and $m \ge d^2$, both the $\ell_p$-ball and $S_q$-ball influence the estimation error, demonstrating an interplay between structured sparsity and low-rank regularization. When $m \le d^2$, the error rate is dictated by $S_q$, making it independent of $m$, emphasizing the fundamental role of rank constraints in this regime.

## 5 Optimization for Dual Spectral Sparse Tensor Estimation

Efficiently solving tensor estimation problems with dual spectral sparsity (**RQ3**) is key to leveraging the proposed $\ell_p$-Schatten-$q$ quasi-norm in practice. However, this task presents substantial challenges due to the non-convexity and coupled structure of this regularization.

**Challenges.** Even in the vector setting, optimizing dual-level sparse structures is notoriously difficult due to the combination of *non-convexity* and *structural coupling* [7, 15]. In our tensor case, these challenges are further compounded by the need to simultaneously enforce inter-frequency sparsity and intra-frequency low-rankness. Most existing tensor optimization methods either treat frequency components independently or impose low-rank constraints without spectral sparsity considerations, making them ill-suited for the proposed dual-spectral regularization. The $\ell_p$-Schatten-$q$ quasi-norm is non-convex whenever $p, q \in (0, 1]$, ruling out standard convex optimization techniques and necessitating a structure-aware, non-convex optimization strategy.

To address these difficulties, our approach is naturally motivated by the structural properties of the problem. We adopt a *proximal update scheme* that takes advantage of the separability of the

---

[2]The conditions in each setting are provided in the appendix.

transform-domain representation $M(\underline{\mathbf{L}})$, allowing frequency-wise updates, along with an iterative reweighting strategy that facilitates optimization in the presence of non-convex regularization.

**Proximal Operator Formulation.** To handle the non-convex $\ell_p$-Schatten-$q$ regularization, we adopt a proximal update scheme that enforces dual spectral sparsity while remaining computationally efficient. Specifically, at iteration $t$, the update is given by solving:

$$\underline{\mathbf{L}}^{t+1} \in \arg\min_{\underline{\mathbf{L}}} \frac{1}{2}\|\underline{\mathbf{L}} - \underline{\mathbf{Z}}\|_{\mathrm{F}}^2 + \lambda \sum\nolimits_{k=1}^{m} \|M(\underline{\mathbf{L}})_{:,:,k}\|_{S_q}^{p/q}, \tag{9}$$

where $\underline{\mathbf{Z}}$ denotes the intermediate variable aggregating previous updates and gradient information. Since the transform $M(\cdot)$ allows slice-wise decomposition [10], Problem (9) reduces to $m$ subproblems over frequency components $k \in [m]$:

$$\min_{\mathbf{A}_k} \frac{1}{2} \|\mathbf{A}_k - M(\underline{\mathbf{Z}})_{:,:,k}\|_{\mathrm{F}}^2 + \lambda \|\mathbf{A}_k\|_{S_q}^{p/q}, \tag{10}$$

where $\mathbf{A}_k := M(\underline{\mathbf{L}})_{:,:,k}$ denotes the $k$-th frontal slice of the transformed tensor $M(\underline{\mathbf{L}})$. Problem (10) is difficult due to the non-convexity and lack of smoothness of the Schatten-$q$ quasi-norm, which admits no closed-form or standard proximal solution in general.

To efficiently approximate Problem (10), we adopt a reweighted $\ell_{1/2}$-surrogate for $\|\mathbf{A}_k\|_{S_q}^{p/q}$ based on singular values:

$$\sum\nolimits_{i=1}^{d} w_{i,k} \cdot \sigma_i(\mathbf{A}_k)^{1/2}, \tag{11}$$

with weights defined as $w_{i,k} = \left( \sum_{j=1}^{d} \varsigma_{j,k}^q + \epsilon \right)^{p/q-1} \cdot \left( \varsigma_{i,k}^{1/2} + \epsilon \right)^{2q-1}$, where $\epsilon$ is a small regularization constant and $\varsigma_{j,k} := \sigma_j(M(\underline{\mathbf{L}}^t)_{:,:,k})$ are the singular values from the previous iterate. The update for each singular value then becomes a soft-thresholding step:

$$\sigma_i^{(t+1)}(M(\underline{\mathbf{L}})_{:,:,k}) = \mathcal{S}_{\lambda w_{i,k}}^{\ell_{1/2}} \left( \sigma_i(M(\underline{\mathbf{Z}})_{:,:,k}) \right), \tag{12}$$

where $\mathcal{S}^{\ell_{1/2}}$ is the proximal operator for the $\ell_{1/2}$-norm (see Appendix for closed-form expression).

After singular value shrinkage, we reconstruct each slice $M(\underline{\mathbf{L}}^{t+1})_{:,:,k} = \mathbf{U}_k \cdot \mathrm{diag}(\boldsymbol{\sigma}^{(t+1)}) \cdot \mathbf{V}_k^\top$, where $\mathbf{U}_k$ and $\mathbf{V}_k$ are from the SVD of $M(\underline{\mathbf{Z}})_{:,:,k}$. Finally, applying the inverse transform yields the updated tensor $\underline{\mathbf{L}}^{t+1}$ in the original domain.

# 6 Experiments

Having established the theoretical foundations and algorithmic framework, we now evaluate the empirical performance of the proposed $\ell_p$-Schatten-$q$ quasi-norm in tensor estimation tasks. We conduct extensive experiments on three types of remote sensing data to demonstrate its effectiveness in noisy tensor completion tasks.

**Experimental Setup.** We consider the noisy tensor completion which involves reconstructing a tensor from noisy incomplete observations. Given a clean tensor $\underline{\mathbf{L}}$ of size $d_1 \times d_2 \times d_3$, we introduce *i.i.d.* Gaussian noise with standard deviation $\sigma = c\sigma_0$, where $c = 0.05$ and $\sigma_0 = \|\underline{\mathbf{L}}\|_{\mathrm{F}}/\sqrt{d_1 d_2 d_3}$. A uniform sampling strategy is applied with sampling ratios $p \in \{0.05, 0.1, 0.15\}$, meaning that 95%, 90%, and 85% of the entries are missing, respectively. Each setting is tested over 10 trials, and the averaged PSNR (dB) and SSIM values are reported. To benchmark our method, we compare the proposed $\ell_p(S_q)$-quasi-norm against several low-rank regularizers, including matrix nuclear norm (NN) [2], Tucker-based tensor nuclear norm (SNN) [16], TNN-DFT [37], TNN-DCT [20], tensor $k$-Support norm ($k$-Supp) ($k = 2$) [29], tensor $\ell_{1-2}$-norm ($\ell_{1-2}$) [26], tensor Schatten-$p$-norm ($p = 1/2$) [12]. In our implementation, we set the sparsity parameters[3] to $(p, q) = (0.8961, 0.8966)$ and employ the Discrete Cosine Transform (DCT) as the transform operator $M(\cdot)$. Details of the experiments are given in the appendix.

---

[3] We first performed a coarse grid search over $p, q \in \{0.1, 0.2, \ldots, 1.0\}$ and observed consistent performance peaks near $p = q = 0.9$. We then manually fine-tuned within $[0.88, 0.92]$ based on PSNR, selecting $(p, q) = (0.8961, 0.8966)$ as the best-performing pair.

Table 1: Results for noisy tensor completion on remote sensing datasets are shown below. The best result in each case is highlighted in **bold**, while the second-best is underlined.

| Dataset | SR | Metric | NN | SNN | TNN-DFT | TNN-DCT | $k$-Supp | $\ell_{1-2}$ | Schatten-1/2 | $\ell_p(S_q)$ (proposed) |
|---|---|---|---|---|---|---|---|---|---|---|
| SalinasA | 5% | PSNR | 15.21 | 20.79 | 22.55 | 26.52 | 22.58 | 22.21 | 22.45 | **28.43** |
| | | SSIM | 0.2594 | **0.7547** | 0.5667 | 0.7384 | 0.5689 | 0.5524 | 0.4474 | 0.7374 |
| | 10% | PSNR | 20.62 | 25.56 | 25.72 | 29.61 | 25.89 | 26.14 | 25.86 | **31.81** |
| | | SSIM | 0.4775 | 0.8284 | 0.7027 | 0.8403 | 0.7231 | 0.7197 | 0.6058 | **0.8484** |
| | 15% | PSNR | 23.09 | 27.99 | 28.06 | 31.32 | 28.09 | 28.13 | 26.98 | **33.23** |
| | | SSIM | 0.5643 | 0.8622 | 0.7804 | 0.8798 | 0.7810 | 0.7795 | 0.6505 | **0.8830** |
| IndianPines | 5% | PSNR | 20.44 | 22.01 | 25.68 | 26.26 | 25.70 | 25.73 | 24.68 | **27.05** |
| | | SSIM | 0.3895 | 0.6359 | 0.6293 | 0.6727 | 0.6289 | 0.6316 | 0.5361 | **0.6740** |
| | 10% | PSNR | 22.23 | 24.94 | 27.45 | 28.40 | 27.48 | 27.52 | 25.72 | **28.92** |
| | | SSIM | 0.4836 | 0.7171 | 0.7226 | **0.7744** | 0.7219 | 0.7249 | 0.5991 | 0.7617 |
| | 15% | PSNR | 23.52 | 26.61 | 28.54 | 29.52 | 28.53 | 28.63 | 26.24 | **29.89** |
| | | SSIM | 0.5438 | 0.7668 | 0.7713 | **0.8177** | 0.7709 | 0.7741 | 0.6258 | 0.7997 |
| Cloth | 5% | PSNR | 20.10 | 20.95 | 25.00 | 26.09 | 25.08 | 25.09 | 24.96 | **26.99** |
| | | SSIM | 0.3762 | 0.5096 | 0.6773 | 0.7283 | 0.6792 | 0.6793 | 0.6305 | **0.7422** |
| | 10% | PSNR | 21.14 | 22.72 | 28.00 | 29.24 | 28.12 | 28.14 | 27.98 | **30.63** |
| | | SSIM | 0.4341 | 0.5983 | 0.8132 | 0.8540 | 0.8143 | 0.8163 | 0.7668 | **0.8658** |
| | 15% | PSNR | 22.05 | 24.18 | 30.03 | 31.36 | 30.08 | 30.11 | 29.50 | **32.71** |
| | | SSIM | 0.4889 | 0.6783 | 0.8722 | 0.9054 | 0.8727 | 0.8733 | 0.8153 | **0.9090** |
| Hair | 5% | PSNR | 25.33 | 30.09 | 33.16 | 35.31 | 33.19 | 33.27 | 33.43 | **36.95** |
| | | SSIM | 0.7147 | 0.8631 | 0.8917 | **0.9248** | 0.8921 | 0.8919 | 0.8240 | 0.9196 |
| | 10% | PSNR | 29.52 | 33.35 | 36.22 | 38.18 | 36.17 | 36.30 | 35.69 | **39.91** |
| | | SSIM | 0.8008 | 0.9122 | 0.9292 | **0.9535** | 0.9286 | 0.9296 | 0.8640 | 0.9517 |
| | 15% | PSNR | 31.12 | 35.24 | 38.00 | 39.88 | 37.91 | 38.07 | 36.46 | **41.52** |
| | | SSIM | 0.8364 | 0.9336 | 0.9449 | **0.9650** | 0.9442 | 0.9448 | 0.8735 | 0.9641 |
| JellyBeans | 5% | PSNR | 16.33 | 18.21 | 25.43 | 26.47 | 25.38 | 25.62 | 25.39 | **27.91** |
| | | SSIM | 0.2397 | 0.4942 | 0.6726 | **0.7223** | 0.6714 | 0.6733 | 0.5504 | 0.7115 |
| | 10% | PSNR | 18.12 | 22.11 | 28.50 | 30.14 | 28.47 | 28.67 | 28.41 | **31.95** |
| | | SSIM | 0.3169 | 0.6629 | 0.7900 | **0.8518** | 0.7902 | 0.7932 | 0.6905 | 0.8486 |
| | 15% | PSNR | 19.92 | 24.67 | 30.51 | 32.33 | 30.52 | 30.61 | 29.96 | **33.97** |
| | | SSIM | 0.4053 | 0.7592 | 0.8489 | **0.9030** | 0.8504 | 0.8499 | 0.7516 | 0.8980 |
| OSU Thermal | 5% | PSNR | 13.19 | 15.83 | 28.06 | 27.99 | 28.01 | 28.19 | 28.11 | **30.06** |
| | | SSIM | 0.1848 | 0.4759 | 0.8584 | 0.8707 | 0.8579 | 0.8603 | 0.7928 | **0.8759** |
| | 10% | PSNR | 14.67 | 19.75 | 31.30 | 31.62 | 31.28 | 31.60 | 30.51 | **33.67** |
| | | SSIM | 0.2509 | 0.6594 | 0.9151 | **0.9326** | 0.9147 | 0.9168 | 0.8358 | 0.9272 |
| | 15% | PSNR | 16.27 | 22.52 | 33.02 | 33.51 | 33.05 | 33.11 | 30.99 | **35.09** |
| | | SSIM | 0.3273 | 0.7621 | 0.9315 | **0.9509** | 0.9321 | 0.9318 | 0.8373 | 0.9404 |

**Datasets.** We validate our approach on three categories of remote sensing data. First, for hyperspectral images, we employ the corrected *Indian Pines* and *Salinas A* datasets from the AVIRIS sensor, containing 200 and 204 spectral bands respectively. Due to computational considerations, we utilize the first 30 bands in our experiments. Second, we evaluate on multispectral images from the Columbia MSI Database, including *Cloth*, *Hair*, and *Jelly Beans*, each with dimensions $512 \times 512 \times 31$ and normalized intensity values in [0,1]. Finally, for thermal imaging, we use sequences from the *OSU Thermal Database*, specifically the first 30 frames of Sequence 1, forming a tensor of size $320 \times 240 \times 30$.

**Results and Analysis.** Table 1 summarizes the PSNR and SSIM results across different missing rates. The proposed $\ell_p(S_q)$-quasi-norm achieves the highest PSNR, demonstrating its effectiveness in preserving spectral information. Its SSIM results rank among the top two, indicating that our approach better retains structural integrity compared to competing methods. These experimental results demonstrate the effectiveness of the proposed $\ell_p$-Schatten-$q$ quasi-norm in robust tensor recovery, showing how characterizing dual spectral sparsity structures in transformed domains benefits tensor reconstruction performance.

# 7 Conclusion

This paper identifies and formalizes a coupled spectral structure within the t-SVD framework, where inter-frequency sparsity coexists with intra-frequency low-rankness. To capture this structure, we propose a unified modeling approach based on the $\ell_p$-Schatten-$q$ quasi-norm, which enables separate control over spectral sparsity at different levels and generalizes existing tensor norms. We provide sharp minimax guarantees under both hard and soft sparsity regimes, and develop an efficient proximal algorithm tailored to this setting. Experimental results demonstrate the practical potential of the proposed approach for structured tensor recovery.

**Limitation.** To highlight the fundamental properties of the proposed $\ell_p$-Schatten-$q$ quasi-norm, our analysis employs several simplifications, including Gaussian location model and idealized sparsity patterns. While our optimization algorithm shows promising empirical performance, its theoretical convergence properties remain to be established. These theoretical and algorithmic limitations suggest important directions for future research.

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

# NeurIPS Paper Checklist

The checklist is designed to encourage best practices for responsible machine learning research, addressing issues of reproducibility, transparency, research ethics, and societal impact. Do not remove the checklist: **The papers not including the checklist will be desk rejected.** The checklist should follow the references and follow the (optional) supplemental material. The checklist does NOT count towards the page limit.

Please read the checklist guidelines carefully for information on how to answer these questions. For each question in the checklist:

- You should answer [Yes] , [No] , or [NA] .
- [NA] means either that the question is Not Applicable for that particular paper or the relevant information is Not Available.
- Please provide a short (1–2 sentence) justification right after your answer (even for NA).

**The checklist answers are an integral part of your paper submission.** They are visible to the reviewers, area chairs, senior area chairs, and ethics reviewers. You will be asked to also include it (after eventual revisions) with the final version of your paper, and its final version will be published with the paper.

The reviewers of your paper will be asked to use the checklist as one of the factors in their evaluation. While "[Yes] " is generally preferable to "[No] ", it is perfectly acceptable to answer "[No] " provided a proper justification is given (e.g., "error bars are not reported because it would be too computationally expensive" or "we were unable to find the license for the dataset we used"). In general, answering "[No] " or "[NA] " is not grounds for rejection. While the questions are phrased in a binary way, we acknowledge that the true answer is often more nuanced, so please just use your best judgment and write a justification to elaborate. All supporting evidence can appear either in the main paper or the supplemental material, provided in appendix. If you answer [Yes] to a question, in the justification please point to the section(s) where related material for the question can be found.

IMPORTANT, please:

- **Delete this instruction block, but keep the section heading "NeurIPS Paper Checklist",**
- **Keep the checklist subsection headings, questions/answers and guidelines below.**
- **Do not modify the questions and only use the provided macros for your answers**.

