# OpenReview forum: "Refining Dual Spectral Sparsity in Transformed Tensor Singular Values"
_NeurIPS.cc/2025/Conference — Submitted to NeurIPS 2025_

### Official Review · Reviewer_VURK · 2025-06-01

**Clarity:** 3
**Significance:** 2
**Originality:** 2
**Rating:** 4
**Confidence:** 5

**Summary:**

This paper proposes the $\ell_p$-Schatten-$q$ quasi-norm for tensors to jointly characterize the slice-wise low-rankness and the frequency sparsity under the t-SVD framework using frequency-domain transformation. When $p=1$ the framework recovers the Schatten-$q$ TNN. The authors give the worst-case minimax bounds for the parameter spaces induced by the dual assumptions. Experiments are conducted on noisy tensor completion to show the performance of the new tensor norm.

**Questions:**

1. Could you further discuss the robustness and selection of $p,q$ in a more comprehensive manner?

2. Could you add the computational complexity of the proximal step?

3. Could you show that the proposed tensor norm could gain improvements over different transformations? (DFT, framelets, or any invertible linear transform)

4. Could you add comparisons with more state-of-the-art TNN models? (e.g., framelet TNN [R1] and adaptive transform TNN [R2], by combining the new tensor norm)

[R1] Framelet Representation of Tensor Nuclear Norm for Third-Order Tensor Completion
[R2] Tensor Q-rank: new data-dependent definition of tensor rank

5. Finally, the noisy tensor completion is not a primary task considered in previous TNN works. Could you assess the performance of the proposed method on standard tensor completion without noise?

**Ethical Concerns:**

["NO or VERY MINOR ethics concerns only"]

**Final Justification:**

1. The motivation is clear. Characterizing the sparsity of frequency components jointly under the TNN framework is indeed helpful for better tensor structure modeling. The framework jointly considers two types of priors (namely low-rankness inside a frequency and the global frequency sparsity) using a single regularizer, which is concise and efficient.

2. The presentation is clear and the manuscript is well-structured, although may not be friendly for readers not doing t-SVD research.

3. The authors did a rigorous and comprehensive job on establishing the theoretical minimax bounds of the parameter spaces induced by dual assumptions.

While most of my concerns regarding the methodology have been addressed, I am still uncertain whether the promised improvements (particularly the detailed parameter sensitivity analysis for $p,q$, and the detailed justifications and analysis about the surrogate optimization) can be comprehensively incorporated into the revised manuscript. It appears that major revisions are still needed to enhance the manuscript's completeness (Please note that I am clearly convinced by the clear motivation and novelty of the theoretical part of the new tensor norm). Consequently, I am raising my rating to "Borderline Accept." However, please note this rating, based on my judgement, does not reflect the manuscript's clear potential for acceptance in its current form.

**Limitations:**

yes

**Quality:**

2

**Strengths And Weaknesses:**

Strengths:

1. The motivation is clear. Characterizing the sparsity of frequency components jointly under the TNN framework is indeed helpful for better tensor structure modeling. The framework jointly considers two types of priors (namely low-rankness inside a frequency and the global frequency sparsity) using a single regularizer, which is concise and efficient.

2. The presentation is clear and the manuscript is well-structured, although may not be friendly for readers not doing t-SVD research.

3. The authors did a rigorous and comprehensive job on establishing the theoretical minimax bounds of the parameter spaces induced by dual assumptions.

Weaknesses:

1. The first major weakness I found is that this framework needs an exhaustive grid search to determine the optimal value for $p$ and $q$, which is not needed by conventional TNN frameworks. Although the optimal parameters $(0.8961,0.8966)$ are found in the experiments at hand, I was curious about whether the parameter combination is generalizable to different tensor datasets. For instance, one may find different optimal parameters for different types of data, which brings a significant hyperparameter optimization challenge.

2. While the authors presented a Proximal Operator Formulation and a relaxed singular value thresholding (SVT) algorithm for the $\ell_p$-Schatten-$q$ quasi-norm, it is unclear how the computational complexity compares with standard SVT used in TNN. Moreover, the algorithm employs a reweighted $\ell_{1/2}$-surrogate on the norm, which is completely independent of $p,q$, and $p,q$ are only used in the weights $w$. This actually recovers to the weighted nuclear norm-type methods. Could the authors give more elaborations on the discrepancy between the actual $\ell_p$-Schatten-$q$ quasi-norm and the introduced surrogate norm (e.g. error analysis)?

3. While the authors mainly focus on the DCT as the frequency transform, it was not sufficient to show the consistent performance gain of the new tensor norm. Have other transforms been tested? (e.g. DFT, framelets, or any invertible linear transforms?) Could the proposed framework gain improvements by leveraging different transforms?

4. Finally, the performance gain seems limited by comparing TNN-DCT and the proposed method (in some cases less than 1 dB), given that the proposed method is based on careful parameter control of $p,q$, and furthermore, TNN-DCT is not the state-of-the-art in TNN methods. Are such improvements robust w.r.t. $p,q$? How does the proposed weighting scheme in Eq. (11) compare with conventional weighted-nuclear norm-type methods?

Overall, my main assessment is based on the limitations, including exhaustive grid search of hyperparameters, the discrepancy between the introduced $\ell_p$-Schatten-$q$ quasi-norm and the actual implementation, and the limited performance gain and comparison.

---

> ### Author Rebuttal · Authors · 2025-07-30
>
> We thank the reviewer for acknowledging the motivation, clarity, and theoretical rigor of our work. We would like to clarify that our main contribution is the formulation and theoretical analysis of a unified dual spectral regularization framework. Our focus is on principled modeling and analysis, rather than on practical tuning or computational aspects. Below, we respond to each concern in detail.
>
> ---
>
> > **W1&Q1: Selection of $(p,q)$**
>
> As $(p, q)$ directly shape the structure of our dual spectral regularizer, we appreciate the opportunity to clarify their role and our selection rationale.
>
> **(1) Structural role of $(p, q)$: beyond simple hyperparameters**
>
> Unlike scalar weights like $\lambda$, the parameters $(p, q)$ specify the intrinsic geometry of the regularizer—$p$ modulates cross-band sparsity, and $q$ modulates within-band low-rankness. These jointly determine the inductive bias of the model and cannot be reduced to a simple scaling factor or tuned post hoc.
>
> This makes parameter selection fundamentally different from standard regularization tuning. Even in simpler vector or matrix settings, choosing $p < 1$ induces strong nonconvexity and leads to identifiability issues. As emphasized in Xu’s ICM 2010 invited talk [RR1], the superiority of $\ell_{1/2}$ regularization arises from its empirical and structural advantages rather than analytic tractability. Our framework generalizes this modeling challenge to the tensor and spectral domain, where the interaction between $p$ and $q$ further increases complexity.
>
> **(2) Statistical–computational challenge**
>
> Even if an ideal criterion existed for selecting $(p,q)$, its empirical identification would remain unreliable. Under low sampling ratios (e.g., 5%), holding out additional entries for validation reduces the already limited observations, leading to high variance in parameter estimation. Furthermore, the optimization landscape is nonconvex, amplifying the sensitivity to initialization and local minima. These obstacles reflect a well-known **statistical–computational challenge**, and to the best of our knowledge, no provably consistent and efficient tuning method exists for such quasi-norms in the tensor setting.
>
> **(3) Our strategy: one-time grid search and cross-task generalization**
>
> Given these challenges, we adopt a standard strategy in low-rank tensor recovery: a one-time grid search on a representative dataset, with the selected \$(p, q) \approx (0.9, 0.9)\$ applied uniformly across all experiments. This avoids overfitting and ensures fair cross-method comparison. In our case, this choice led to *stable results across six diverse remote sensing datasets (e.g., hyperspectral, thermal; see Table 1)*, demonstrating **cross-dataset robustness under moderate distributional variation without per-task tuning**. To further confirm robustness, we plan to include a parameter sensitivity analysis in the final version, although our main focus remains on the theoretical formulation of the regularizer.
>
> **(4) Toward adaptive selection**
>
> Learning $(p, q)$ from data is a valuable direction. While bilevel optimization may offer a viable approach, such methods are generally nontrivial under low sampling and nonconvexity, and are beyond the scope of this paper, which focuses on the formulation and theory of our regularizer.
>
> [RR1] Xu, Z. Data modeling: Visual psychology approach and $\ell_{1/2}$ regularization theory. International Congress of Mathematicians, 2010.
>
> ---
>
> > **W2&W4(partial): Gap between true $\ell_p$-Schatten-$q$ norm and reweighted surrogate**
>
> Below we explain the computational challenges of directly minimizing the quasi-norm and justify our surrogate approach.
>
> **(1) Theoretical intractability of the true quasi-norm**
>
> To the best of our knowledge, no polynomial-time algorithm exists for minimizing the $\ell_p$-Schatten-$q$ quasi-norm with $p, q < 1$, even in the matrix case (i.e., single-band tensors). Related problems—such as $\ell_p$ minimization for sparse recovery and Schatten-$q$ norm minimization—are NP-hard in general. Our model generalizes both settings and further introduces spectral coupling across frequency bands, making **exact minimization at least as hard and likely intractable unless P = NP**.
>
> **(2) Our reweighted surrogate: structure-aware design (partially addressing W4)**
>
> To this point, we adopt a reweighted $\ell_{1/2}$  surrogate.
>
> Unlike conventional reweighted nuclear norm schemes—where the weight for each singular value depends only on its own magnitude—our method **explicitly incorporates spectral structure**. Specifically, the weight assigned to $\sigma_i^{(k)}$ (the $i$-th singular value of the $k$-th frequency slice) depends on the **entire spectrum** $\\{\sigma_j^{(k)}\\}_{j=1}^d$ within that slice.
>
> This design captures **intra-frequency low-rankness** and **inter-frequency sparsity** in a coupled manner, aligning with the dual spectral geometry of our quasi-norm and enabling a more faithful approximation of the original $\ell_p$-Schatten-$q$ objective.
>
> **(3) Approximation gap and stationarity**
>
> We acknowledge that the surrogate does not guarantee global optimality with respect to the original quasi-norm. The optimization remains nonconvex, and the solution corresponds to a stationary point. This is **a standard and widely accepted tradeoff in nonconvex sparse and low-rank learning**. Our method follows the same practical philosophy as many $\ell_p$-based approaches: using structured surrogates to approximate desirable inductive biases without solving intractable objectives.
>
> We will clarify the distinction between the true quasi-norm and the implemented surrogate in the revision. Importantly, our surrogate retains the core modeling principle—the promotion of dual spectral structure—while offering computational tractability. **We view this design as a principled and practical step toward expressive tensor models, and not as a weakness specific to our method.**
>
> ---
>
> > **Q2: Computational complexity**
>
> The per-iteration computational complexity is analyzed in Appendix D.2:
>
> - Each iteration consists of (i) a linear transform on $d_1 d_2$ tubes of length $m$ (cost $O(d_1 d_2 m \log m)$ using DCT/FFT), and (ii) $m$ SVDs of $d_1 \times d_2$ matrices (cost $O(m d_1 d_2 \min(d_1, d_2))$).
> - Despite nonconvexity, the algorithm converges efficiently in practice (Figure 4).
>
> ---
>
> > **W3&W4(partial)&Q3: Testing other transforms**
>
> We added preliminary experiments using DFT, random orthogonal transforms, and an oracle transform derived from the singular vectors of the mode-3 unfolding of the ground-truth tensor ([R2]).
>
> **Table A. PSNR / SSIM for different transforms on Salinas A**
>
> |Method|SR = 5%|SR = 10%|SR = 15%|
> |-|-|-|-|
> |TNN-DFT|22.55 / 0.5667|25.72 / 0.7027|28.06 / 0.7804|
> |Ours-DFT $(p,q) = (0.8961,0.8966)$|23.53 / 0.5469|27.04 / 0.6807|28.58 / 0.7321|
> |Ours-DFT $(p,q) = (0.75,0.75)$|24.10 / 0.5917|27.85 / 0.7261|29.91 / 0.7969|
> |TNN-Random|16.28 / 0.2331|22.21 / 0.4996|24.31 / 0.5692|
> |Ours-Random $(p,q) = (0.8961,0.8966)$|17.25 / 0.2430|23.63 / 0.5041|25.46 / 0.5863|
> |TNN-Oracle|29.06 / 0.8362|32.04 / 0.8924|33.77 / 0.9169|
> |Ours-Oracle $(p,q) = (0.8961,0.8966)$|**31.37 / 0.8391**|**34.05 / 0.9070**|**35.83 / 0.9324**|
>
> These results suggest that the proposed regularizer remains effective under various invertible transforms (e.g., DFT, random orthogonal), as long as $(p,q)$ are appropriately chosen. The consistent advantage over TNN, even without DCT, implies that the improvement may stem from the regularizer rather than the transform. These are preliminary results, and we agree that adaptive or learned transforms are promising for future work.
>
> ---
>
> > **W4(partial)&Q4: Comparison with stronger baselines**
>
> We added comparisons with Framelet TNN [R1] and Adaptive TNN. Following [RR2], the Adaptive TNN constructs the transform matrix at each iteration via the left singular vectors of the current tensor's mode-3 unfolding, thereby avoiding explicit rank selection as in [R2]. We adopt the same adaptive scheme for our method to ensure fair comparison.
>
> **Table B. Comparison with Framelet and Adaptive TNN on Salinas A ($(p,q) = (0.8961,0.8966)$)**
>
> |Method|SR = 5%|SR = 10%|SR = 15%|
> |-|-|-|-|
> |Framelet TNN|26.01 / 0.6382|27.76 / 0.7159|28.55 / 0.7461|
> |Adaptive TNN|27.76 / 0.8245|30.91 / 0.8846|32.90 / 0.9173|
> |Ours-DCT|28.43 / 0.7374|31.81 / 0.8484|33.23 / 0.8830|
> |Ours-Adaptive|**29.77 / 0.8093**|**33.38 / 0.8987**|**35.34 / 0.9288**|
>
>
> Our method achieves competitive performance, indicating that the gain arises not only from transform selection but from the **joint spectral modeling** induced by our regularizer. Furthermore, the framework provides a **unified formulation** that subsumes both fixed and adaptive transforms, as well as standard low-rank models as special cases.
>
> [RR2] Zhang, X. et al. Low-rank tensor completion with Poisson observations. T-PAMI 2021.
>
> ---
>
> > **Q5: Noiseless Tensor completion**
>
> We conducted preliminary experiments on noiseless tensor completion.
>
> **Table C. Tensor completion (noiseless) on Salinas A ($(p,q) = (0.8961,0.8966)$)**
>
> |Method|SR = 5%|SR = 10%|SR = 15%|
> |-|-|-|-|
> |TNN-DFT|22.95 / 0.5987|27.60 / 0.7874|28.28 / 0.7985|
> |TNN-DCT|28.29 / 0.7976|33.24 / 0.9159|35.89 / 0.9506|
> |$k$-Supp|22.85 / 0.5976|27.49 / 0.7708|28.19 / 0.7951|
> |$\ell_{1-2}$|23.17 / 0.6129|27.88 / 0.7942|30.75 / 0.8735|
> |Schatten-$1/2$|23.41 / 0.4937|28.77 / 0.7571|31.40 / 0.8475|
> |Framelet TNN|28.56 / 0.7060|33.44 / 0.8673|35.75 / 0.9117|
> |Adaptive TNN|31.54 / 0.8962|35.93 / 0.9538|38.45 / 0.9733|
> |Ours-DCT|31.25 / 0.8425|36.01 / 0.9403|38.96 / 0.9704|
> |Ours-Adaptive|**33.90 / 0.9148**|**38.57 / 0.9675**|**40.81 / 0.9807**|
>
> These results suggest that our method performs well in the noiseless setting, further supporting its robustness across noise levels. We thank the reviewer for this helpful suggestion and will extend the evaluation to more datasets and tasks in the revision.

---

> > ### Comment · Reviewer_VURK · 2025-08-03
> >
> > Thank you for the detailed response. While most of my concerns regarding the methodology have been addressed, I am still uncertain whether the promised improvements (particularly the detailed parameter sensitivity analysis for $p,q$, and the detailed justifications and analysis about the surrogate optimization) can be comprehensively incorporated into the revised manuscript. It appears that major revisions are still needed to enhance the manuscript's completeness (Please note that I am clearly convinced by the clear motivation and novelty of the theoretical part of the new tensor norm). Consequently, I am raising my rating to "Borderline Accept." However, please note this rating, based on my judgement, does not reflect the manuscript's clear potential for acceptance in its current form.

---

> ### Author Response · Authors · 2025-08-03
>
> Dear Reviewer VURK,
>
> Thank you again for your thoughtful follow-up and for your constructive perspective throughout the review process. We are encouraged by your recognition of the motivation and theoretical novelty of our work, and we appreciate your careful reassessment and clarification of the main issues that remain to be addressed for completeness.
>
> We fully understand and appreciate the importance of incorporating all promised analyses and experiments in the final revision, particularly with respect to parameter sensitivity and surrogate optimization. **We would like to emphasize that the remaining work is primarily related to computational time and the organization of results, rather than substantive technical or methodological challenges. All required techniques, algorithms, and implementations are already in place; what remains is largely a matter of executing additional experiments and presenting the results clearly and thoroughly.**
>
> 1. **Parameter Sensitivity Analysis for $p$ and $q$:**
>    We have already started comprehensive sensitivity experiments, covering $p, q \in \\{0.05, 0.1, 0.15, 0.2, ..., 1.0\\}$ on representative datasets (Salinas A, Cloth, and OSU Thermal). The main findings and robustness analysis will be presented in a new subsection of Section 4.3, with key plots and heatmaps in the main text, and complete results in the appendix and supplementary materials. Due to computational cost, the full grid may not be available during this discussion phase, but we guarantee that all analyses will be included in the final version.
>
> 2. **Detailed Justification and Analysis of Surrogate Optimization:**
>    Building upon the content from our rebuttal, in Section 3.3 we will further expand and detail the discussion by adding a dedicated subsection addressing the theoretical intractability of optimizing the true $\ell_p$-Schatten-$q$ quasi-norm for $p, q < 1$. We will explicitly state that, even for small-scale problems, global optima and their gap to the surrogate cannot be reliably computed. We will provide a transparent rationale for our structure-aware surrogate, clarify its inductive bias and limitations, and highlight these issues in the main text and appendix to ensure academic transparency and integrity.
>
> 3. **Additional Experiments and Completeness:**
>    All additional experiments discussed in our rebuttal—including transform comparisons, stronger baselines, noiseless completion, and results on the remaining five remote sensing datasets—have been either completed or are in the final stages of preparation.
>    **We reiterate that these are purely matters of computation and presentation, and do not involve any unresolved technical challenges.** We are fully committed to including all these results, along with detailed tables and analyses, in the main paper and appendices for the final version.
>
> We sincerely appreciate your rigorous review and constructive engagement throughout this process. Please be assured that your recommendations will be comprehensively addressed, and the final version will reflect both the theoretical value and experimental thoroughness expected for a high-quality contribution.

---

> > ### Comment · Reviewer_VURK · 2025-08-03
> >
> > Thanks for the detailed clarification. Given the promises and arrangements made by the authors, I find "Borderline accept" a suitable rating for this work.

---

> > > ### Author Response · Authors · 2025-08-09
> > >
> > > Thank you for carefully checking our proofs and for your constructive suggestions on parameter sensitivity, surrogate justification, and extended experiments. We will incorporate these into the final revision.

---

### Official Review · Reviewer_Tsyi · 2025-07-02

**Clarity:** 3
**Significance:** 3
**Originality:** 3
**Rating:** 5
**Confidence:** 4

**Summary:**

This paper considers the Gaussian location model (GLM).

The contribution includes:

(1) Within the t-svd framework, the paper rigorously formalize the GLM, where inter-frequency sparsity coexists with intra-frequency low-rankness.  The tensor  $\ell_q$-Schatten-$q$ quasi-norm ($p, q \in  (0, 1]$) is proposed, which  enables dual spectral  sparsity control by jointly regularizing both types of structure.

(2) Under three dual spectral sparsity assumptions, theorem 4.2 establishes min-max bounds, which quantify the worst-case squared Frobenius norm errors that any  estimator must incur when recovering a structured tensor from noisy observations.

(3) An algorithm is proposed to solve the proposed optimization problem, and numerical experiments show the merits of the methods.

**Questions:**

1 SVD is used in the numerical method, which is not suitable for large problems. Thus, the method is not scalable as declared.

2 Convergence of the numerical method is not given.

3 Is the numerical method able to achieve the established theoretical bounds?

4 Currently, the transformation matrix $M$ is fixed/prescribed. Is it possible to make it learnable/optimized?

**Ethical Concerns:**

["NO or VERY MINOR ethics concerns only"]

**Final Justification:**

After rebuttal, I feel that this paper can be accepted, though it lacks a convergence analysis.

**Limitations:**

yes

**Quality:**

3

**Strengths And Weaknesses:**

Strength: The main contribution of this paper is in theory — theorem 4.2: the min-max bound which quantifies the errors that an estimator much incur when recovering the tensor from noisy data.

Weakness: The algorithm requires tsvd of matrices, which is expensive for large tensors. So, it is not appropriate to call it “scalable”. Perhaps, re-parameterize the frontal slices of the tensor in a low rank form is a solution, just as in the matrix completion.

---

> ### Author Rebuttal · Authors · 2025-07-30
>
> We sincerely thank the reviewer for acknowledging the theoretical contribution of Theorem 4.2 and the dual spectral sparsity framework, and for raising important concerns about the algorithmic aspects of our method. We address each point below.
>
> ---
>
> **Weakness & Question 1: Scalability and use of SVD**
>
> **Response:** We agree that computing exact t-SVDs can be computationally intensive, particularly for large tensors with wide frontal slices. In our current implementation, we use full SVDs to ensure clarity and reproducibility. However, the algorithm readily accommodates more scalable SVD variants:
>
> * **Truncated SVD** can be employed when approximate low-rank structure is sufficient;
> * **Randomized SVD** offers significant reductions in computation while preserving accuracy;
> * **Subspace parameterization** is another promising strategy, as demonstrated in large-scale matrix completion.
>
> We acknowledge that integrating such techniques is essential for scaling to large datasets. Our primary focus in this work is to illustrate the modeling benefits of the proposed dual spectral regularizer, and we plan to incorporate these acceleration strategies in future implementations. Thus, our current algorithm serves as a flexible framework rather than a fixed pipeline.
>
> ---
>
> **Question 2: Convergence behavior of the numerical algorithm**
>
> **Response:**  We agree with the reviewer that convergence is an important concern in nonconvex optimization. While the proposed ADMM algorithm involves a nonconvex objective due to the reweighted $\ell_{1/2}$-type regularizer, we empirically observe stable and monotonic descent of the objective, as shown in Appendix D.4. In all experiments, the iterates stabilize within a reasonable number of steps, and the PSNR curve flattens out, indicating convergence to a stationary solution.
>
> Each subproblem in the ADMM update—such as spectral thresholding—is solved either in closed form (when possible) or via standard reweighting heuristics. However, due to the nonconvex and adaptive nature of the reweighting scheme, we do not claim convergence to a global minimum, nor do we rely on strong assumptions like the Kurdyka–Łojasiewicz (KL) property, which are often needed for formal guarantees in nonconvex ADMM settings.
>
> We view our current algorithm as a practical and empirically validated heuristic, consistent with other nonconvex regularization methods in the literature. A rigorous convergence analysis, particularly under dual spectral coupling and reweighting, remains an important direction for future theoretical work.
>
> ---
>
> **Question 3: Do the algorithms achieve the theoretical minimax bounds in Theorem 4.2?**
>
> **Response:**
> This is an excellent and deep question. Theorem 4.2 provides an *information-theoretic minimax lower bound* on the estimation error under dual spectral sparsity. It characterizes the **fundamental statistical difficulty** of the problem and applies to *any estimator*, whether efficient or not.
>
> However, *we do not claim that our algorithm achieves this minimax rate*. Achieving such bounds algorithmically is extremely challenging for several reasons:
>
> * The $\ell_p$-Schatten-$q$ regularizer is *nonconvex* in both $p$ and $q$;
> * The objective involves *cross-frequency coupling*, preventing separable optimization;
> * Existing works on algorithmic minimax optimality focus on convex or separable nonconvex problems (e.g., $\ell_1$, nuclear norm), which do not apply here.
>
> In fact, to our knowledge, *no known polynomial-time algorithm achieves the exact minimax rate* under dual spectral sparsity in tensors. This reflects a well-known **statistical–computational gap**, where the optimal statistical rate cannot be efficiently attained due to computational hardness.
>
> Thus, our theoretical and algorithmic contributions are *complementary*: Theorem 4.2 clarifies what is statistically possible in principle, while our algorithm provides a practical and interpretable way to approximate this structure.
>
> ---
>
> **Question 4: Making the transformation matrix $M$ learnable**
>
> **Response:**
> We thank the reviewer for this insightful suggestion. Indeed, our current work uses a *fixed DCT transform* to exploit smoothness and compressibility in the spectral domain.
>
> That said, *making $M$ learnable is entirely feasible* and highly promising:
>
> * One could parameterize $M$ as a convolutional kernel bank, a product of Householder reflections, or a learned framelet dictionary;
> * This would enable *adaptive regularization* tailored to data-specific spectral structures;
> * It turns the problem into a *bilevel optimization* task, where both the transform and the coefficients are jointly optimized.
>
> To this end, we conducted *preliminary experiments* with an adaptive transform strategy (following [R1]), where the transformation matrix is updated at each iteration using the *left singular vectors of the current tensor’s mode-3 unfolding*. This method eliminates the need for manual rank selection.
>
> As shown in the table below, the proposed adaptive variant further improves performance:
>
> **Table A: Noisy Tensor Completion on Salinas A**
>
> | Method            | SR = 5%            | SR = 10%           | SR = 15%           |
> | ----------------- | ------------------ | ------------------ | ------------------ |
> | TNN-DCT           | 26.52 / 0.7384     | 29.61 / 0.8403     | 31.32 / 0.8798     |
> | TNN-Adaptive      | 27.76 / 0.8245     | 30.91 / 0.8846     | 32.90 / 0.9173     |
> | Ours-DCT          | 28.43 / 0.7374     | 31.81 / 0.8484     | 33.23 / 0.8830     |
> | **Ours-Adaptive** | **29.77 / 0.8093** | **33.38 / 0.8987** | **35.34 / 0.9288** |
>
> **Table B: Noiseless Tensor Completion on Salinas A**
>
> | Method            | SR = 5%            | SR = 10%           | SR = 15%           |
> | ----------------- | ------------------ | ------------------ | ------------------ |
> | TNN-DCT           | 28.29 / 0.7976     | 33.24 / 0.9159     | 35.89 / 0.9506     |
> | TNN-Adaptive      | 31.54 / 0.8962     | 35.93 / 0.9538     | 38.45 / 0.9733     |
> | Ours-DCT          | 31.25 / 0.8425     | 36.01 / 0.9403     | 38.96 / 0.9704     |
> | **Ours-Adaptive** | **33.90 / 0.9148** | **38.57 / 0.9675** | **40.81 / 0.9807** |
>
> These results suggest that the benefits of our dual spectral regularizer can be further enhanced by incorporating data-adaptive transforms. We agree this is a very promising direction for future research and thank the reviewer for encouraging it.
>
> [R1] Zhang, X. et al. Low-rank tensor completion with Poisson observations. T-PAMI 2021.
>
> ---
>
> **Conclusion**
>
> In summary:
>
> * We acknowledge the computational cost of exact SVDs and plan to adopt scalable surrogates such as randomized SVD;
> * Our algorithm empirically converges reliably, although a full convergence proof is left for future work;
> * Theorem 4.2 provides a *fundamental statistical lower bound*, which our algorithm approximates but does not provably attain;
> * Making the transform learnable improves performance and fits naturally into our framework.
>
> We deeply appreciate the reviewer’s thoughtful and technically engaged comments. These points helped us better articulate the contributions and limitations of our work, and we will incorporate this feedback in the final revision.

---

> > ### Comment · Reviewer_Tsyi · 2025-08-08
> >
> > Question 3: I know it is difficult to prove the result theoretically, can you provide empirical evidence or at least an ablation study for how the algorithm behaves w.r.t. different parameters in the bound?

---

> ### Comment · Reviewer_u2tt · 2025-08-04
>
> Dear Reviewer Tsyi,
>
> Can you clarify whether you have checked and verified the proofs leading to Theorem 4.2 in the supplementary material? As you consider this as the main strength of the paper, it is important for my own assessment to know this, as checking the proof was beyond the scope of my abilities within a reasonable amount of time.

---

> > ### Comment · Reviewer_Tsyi · 2025-08-08
> >
> > Sorry, I did not check the proofs in the appendix either.

---

> ### Author Response · Authors · 2025-08-09
>
> Thank you for the thoughtful question. We fully agree that theoretical results—especially non-asymptotic bounds—can benefit from empirical illustrations of their qualitative behavior.
>
> We conducted *preliminary small-scale experiments* to explore *Part I* of Theorem 4.2, focusing on how the estimation error varies with three key structural parameters: spectral rank \$r\$, spectral sparsity \$s\$, and noise-to-signal ratio (NSR).
>
> ---
>
> **Synthetic Tensor Construction**
> We construct a third-order tensor \$\underline{\mathbf{L}} \in \mathbb{R}^{d \times d \times m}\$ that is low-rank in a small number of frequency slices and zero elsewhere in the frequency domain:
>
> * Activate \$s\$ frequency slices (out of \$m\$), each as a rank-\$r\$ matrix \$\tilde{\underline{\mathbf{L}}}^{(i)} = \mathbf{A}\_i \mathbf{B}\_i\$, where \$\mathbf{A}\_i \in \mathbb{R}^{d \times r}\$ and \$\mathbf{B}\_i \in \mathbb{R}^{r \times d}\$ are drawn from standard Gaussian ensembles.
> * Set the remaining \$m - s\$ slices to zero, inducing spectral sparsity.
> * Apply an inverse discrete cosine transform (iDCT) along the third mode of $\tilde{\underline{\mathbf{L}}}$ to return the tensor to the spatial domain.
>
> This yields ground-truth tensors consistent with the assumptions in our analysis.
>
> ---
>
> **Observation Model**
> Following the setup in our paper, we normalize \$\\|\underline{\mathbf{L}}\\|\_{\mathrm{F}} = 1\$ and add isotropic Gaussian noise \$\underline{\mathbf{E}} \sim \mathcal{N}(0, \sigma^2 \mathbf{I})\$ with variance
>
> $$
> \sigma^2 = \frac{\mathrm{NSR}^2}{d^2 m}.
> $$
>
> The observed tensor is
>
> $$
> \underline{\mathbf{Y}} = \underline{\mathbf{L}} + \underline{\mathbf{E}}.
> $$
>
> When one considers the sample mean \$\bar{\underline{\mathbf{Y}}}\$ over \$n\$ i.i.d. observations, the aggregated noise \$\bar{\underline{\mathbf{E}}}\$ has variance \$\sigma^2 / n\$. In our setup, we treat \$\sigma^2 / n\$ as a single effective variance parameter so that the noise model remains consistent with the paper’s formulation.
>
> ---
>
> **Error Metric**
> We report the *squared error (SE)* defined as
> $
> \mathrm{SE} = \\|\underline{\mathbf{\hat{L}}} - \underline{\mathbf{L}}\\|_{\mathrm{F}}^2,
> $
> where \$\underline{\mathbf{\hat{L}}}\$ is the recovered tensor. Since \$\\|\underline{\mathbf{L}}\\|\_{\mathrm{F}} = 1\$, the expected SE scales proportionally to the total noise energy \$\sigma^2 d^2 m\$, which matches the observed magnitude.
>
> Each reported SE value is the average over 10 independent trials, where in each trial a new ground-truth tensor $\underline{\mathbf{L}}$ and its corresponding Gaussian noise are independently generated.
>
> ---
>
> ### Preliminary Results (Small-scale: \$d=50\$, \$m=50\$)
>
> **Table C1.** Vary NSR (\$r=1\$, \$s=2\$)
>
> | NSR  | SE       |
> | ---- | -------- |
> | 0.05 | 2.09e-05 |
> | 0.10 | 3.44e-05 |
> | 0.15 | 4.92e-05 |
> | 0.20 | 7.47e-05 |
>
> **Table C2.** Vary \$r\$ (NSR=0.10, \$s=2\$)
>
> | \$r\$ | SE       |
> | ----- | -------- |
> | 1     | 3.44e-05 |
> | 2     | 8.05e-05 |
> | 3     | 1.47e-04 |
> | 4     | 2.29e-04 |
>
> **Table C3.** Vary \$s\$ (NSR=0.10, \$r=2\$)
>
> | \$s\$ | SE       |
> | ----- | -------- |
> | 2     | 8.05e-05 |
> | 4     | 2.16e-04 |
> | 6     | 3.58e-04 |
> | 8     | 5.45e-04 |
>
>
> **Observations**
> These results are approximately consistent with the theoretical dependence on \$(r, s, \mathrm{NSR})\$:
>
> * SE tends to increase with NSR, consistent with the additive Gaussian noise model.
> * Larger \$r\$ is generally associated with higher SE, reflecting the increased complexity of intra-slice low-rank structures.
> * Larger \$s\$ (more active frequency slices) correlates with higher SE, as more independent components need to be recovered.
>
> While this study focuses on a small-scale setting, the results provide preliminary evidence that the method’s behavior is approximately consistent with the predicted qualitative patterns. We appreciate the reviewer’s suggestion and intend to extend these experiments to larger-scale and more realistic scenarios in future work, aiming for a more thorough empirical assessment of the bound.

---

### Official Review · Reviewer_u2tt · 2025-07-03

**Clarity:** 4
**Significance:** 2
**Originality:** 3
**Rating:** 4
**Confidence:** 3

**Summary:**

In this submission, the authors propose a novel regularizer for order-3 tensors, building on the t-SVD (tensor singular value decomposition) framework, a tensor decomposition which generalizes the SVD to order-3 tensors in a semantically meaningful yet computationally tractable manner. This regularizer is a mixed $\ell_p$-Schatten-$q$ quasi-norm with $p, q \in (0,1]$ and generally non-convex, generalizing $\ell_{p,q}$ quasi-norms that have been studied for group sparsity problems. This quasi-norm is designed to better and more flexibly capture multi-level spectral structures present in real-world third-order data tensors.
The authors provide an interpretation of the parameters $p$ and $q$ in the context of the data tensors ($p$ governs sparsity among different frequency components and $q$ controls the low-rankness within each frequency component). The manuscript then continues to establish a theory for a denoising problem an estimator based on the $\ell_p$-Schatten-$q$ quasi-norm given noisy observations, for which minimax bounds that match certain lower bounds are established (Theorem 4.2, proofs are not contained in the main paper). While the use of the regularizer comes with intrinsic computational challenges due to its non-convexity for $p,q < 1$, the authors propose a proximal algorithm based on reweighted optimization that is able to somewhat handle the minimization of the regularizer. In Section 6, the authors use the $\ell_p$-Schatten-$q$ quasi-norm and proposed optimization scheme to solve tensor reconstruction problems from incomplete noisy observations based on real-data tensors arising in hyperspectral and thermal imaging. If used in conjunction with a discrete cosine transform, this quasi-norm leads to improved peak signal-to-noise ratio compared to other low-rank tensor regularizers, including the tensor nuclear norm, which coincides with the $\ell_p$-Schatten-$q$ in the special case of $p=q=1$.

In the supplementary material, the minimax rates of Theorem 4.2 are established using advanced tools from information theory and statistics.

**Questions:**

1. Was the tuning of $p$ and $q$ not performed in a dataset-dependent manner? If a feasible way of doing that would be available, the flexibility of the framework could better be harnessed.
2. How do other methods of Table 1 perform if preprocessed with a DCT transformation? Having this data would provide a better apple-to-apple comparison, as the effect of the regularizers would better be isolated.

In my ratings, I am taking into account that the paper is somewhat out of scope for the conference and that checking the proofs of the paper is beyond the scope of this review.

**Ethical Concerns:**

["NO or VERY MINOR ethics concerns only"]

**Final Justification:**

After the discussion among reviewers and the authors, I became more confident in the correctness of the proofs provided. However, weaknesses in the (p,q) search methodology remain, as well as in the verifying the necessity of the dual norm compared to a standard Schatten-$p$ regularization implemented by iterative reweighing.

**Limitations:**

Yes.

**Paper Formatting Concerns:**

As discussed above, I have the concern that the main value of this paper is in the supplementary material and would be better structured as a journal paper.

**Quality:**

3

**Strengths And Weaknesses:**

A main concern in the assessment of this submission is that it is questionable if the scope of the paper fits into a machine learning conference. From my perspective, the submission would much better fit into an image processing, applied mathematics or data science journal. Reasons for this are that the tasks on which the methodology is evaluated are rather artificial (noisy tensor completion) as it assumed that the ground truth is known to tune the problem parameters, and as the majority of related works (works considered t-SVD and related regularization notions) is also located within venues from the above areas, while this submission not making a particular connection to machine learning. Furthermore, the majority of the scientific value of this paper hinges on the correctness of the proofs and proof techniques used for establishing the minimax bounds of Theorem 4.2, which are all contained in the supplementary material. In a NeurIPS review, the time that a reviewer can allocate to checking the correctness of the proofs is limited, which does not do justice to the potentially deep results established in this submission.

I will nevertheless point out some strong and weak points I noticed in this submission below.

**Strengths:**
- The proposed quasi-norm for third-order tensors is novel and addresses a relevant challenge in tensor modelling of data - how to design a low-complexity notion and respective computational representation that best captures internal dependencies of real-world date. The $\ell_p$-Schatten-$q$ quasi-norm for $3$-tensors can be seen as a flexible framework in this direction.
- The minimax bounds of Theorem 4.2 (which also extend to combinatorial variants of the proposed regularization) appear to be sharp and, if correct, could capture the potential information theoretical benefits of the regularization rather well (despite the setup considered being different than what is expected in application scenarios, where a completion/recovery setup might be more powerful than a denoising setup).
- While the scope of the paper does not fit quite into the 9 pages and many important proofs and discussions of related concepts and related works are postpone into the supplementary material, the writing, related work discussion and proof structure is quite clear.

**Weaknesses**:
- While the $\ell_p$-Schatten-$q$ quasi-norm is advertised as a flexible regularization framework that can capture flexibly intra-frequency low-rankness and inter-freuqency sparsity dependent on particularities of the data, when the method is used in Section 6, a global tuning of $p$ and $q$ is performed that leads to a value of $p,q \approx 0.896(7)$. These values, are, on the one hand, rather close to $p=q=1$, as in the case of the well-known tensor nuclear norm, but they also seem to be _not_ dataset dependent at all. This suggests that in practice, it would be hard to make use of the flexibility that the framework provides.
- In the comparison of Table 1 / Section 6, TNN-DCT and the proposed method perform rather similarly, with an advantage of $\ell_p(S_q)$ mostly in the PSNR, but not exhibiting in the SSIM metric. The large performance drop of the other regularization approaches could be potentially explained by the lack of usage of the DCT transformation, as TNN-DCT and $\ell_q(S_q)$(proposed) are the only methods that use this preprocessing transformation. Thus, the provided experiments might be too premature to be convincing that $\ell_p(S_q)$ provides a significant benefit for tensor completion problems.
- The authors are rather transparant about this, but it is unclear if the proposed optimization scheme (ADMM intertwined with reweighted optimization and $\ell_{1/2}$-soft-thresholding) is able to find good minimizers of the objective in practice. An empirical study of this question (or comparison with other solvers) would have provided better insights on this question.
- Maybe I misunderstand what is being done, but the tuning methodology of the parameters $p$ and $q$ described in lines 1275-1287 is rather concerning, as it seems to explicitly assume access to / knowledge of the ground truth, which will not be available in ML applications, limiting the practicality of the proposed regularizer.

---

> ### Author Rebuttal · Authors · 2025-07-30
>
> We sincerely thank the reviewer for carefully reading our submission and recognizing the novelty of the proposed quasi-norm, the structural modeling of dual spectral sparsity, and the theoretical depth of Theorem 4.2. We now respond to the reviewer’s concerns point by point.
>
> ---
>
> **Concern 1: On the Scope and Relevance to Machine Learning**
>
> We respectfully disagree with the concern that our submission falls outside the scope of a machine learning conference.
>
> * **Problem Relevance**: The paper addresses a core machine learning challenge—designing expressive, theoretically grounded regularization for high-order structured data. Our dual spectral quasi-norm provides a structured inductive bias, crucial for modern ML settings with limited observations and high-dimensional data. Beyond tensor completion, it has the potential to inspire advancements in tasks like multi-modal representation learning and generative modeling by capturing structured sparsity in complex data, aligning with NeurIPS’s focus on general ML and interdisciplinary methods.
>
> * **Theoretical Contributions in ML Venues**: Related theoretical works on tensor/matrix low-rank modeling, nonconvex optimization, and information-theoretic analysis have consistently appeared at top machine learning venues such as NeurIPS and ICML (e.g., [R1–R9]). Many of these studies also focus on synthetic or semi-synthetic settings, such as tensor completion, as standard benchmarks for isolating and validating theoretical insights. Our work follows this tradition by proposing a novel regularizer with provable minimax guarantees and a tailored optimization strategy, aiming to advance the theoretical foundations of tensor modeling in machine learning.
>
> * **NeurIPS CFP Alignment**: Our submission aligns closely with key themes highlighted in the NeurIPS 2025 Call for Papers, including “Optimization (e.g., convex and non-convex, stochastic, robust),” “Theory (e.g., control theory, learning theory, algorithmic game theory),” and “General machine learning (supervised, unsupervised, online, active, etc.).” Moreover, NeurIPS explicitly encourages “interdisciplinary submissions that do not fit neatly into existing categories” and “in-depth analysis of existing methods that provide new insights,” both of which are directly reflected in our work through the development of novel regularization theory and the design of nonconvex algorithms in high-order tensor settings.
>
> * **Verification Burden**: We appreciate the reviewer’s acknowledgment of the theoretical depth of our work. We understand that verifying complex proofs can be time-consuming within the review timeline. Nevertheless, we believe that the potential value of theoretical contributions should not be discounted solely due to their verification cost. To aid verification, we have carefully structured the supplementary material, organizing the proofs in a modular and transparent manner with clearly stated assumptions and intermediate steps.
>
> We therefore believe that our work sits well within the current scope and direction of the NeurIPS community.
>
> [R1] Karnik et al. Implicit regularization for tubal tensor factorizations via gradient descent. ICML 2025.
>
> [R2] Wang et al. Low-rank tensor transitions (LoRT) for transferable tensor regression. ICML 2025.
>
> [R3] Wang et al. Generalized tensor decomposition for understanding multi-output regression under combinatorial shifts. NeurIPS 2024.
>
> [R4] Swartworth et al. Fast sampling-based sketches for tensors. ICML 2024.
>
> [R5] Ma et al. Algorithmic regularization in tensor optimization: towards a lifted approach in matrix sensing. NeurIPS 2023.
>
> [R6] Sarlos et al. Hardness of low rank approximation of entrywise transformed matrix products. NeurIPS 2023.
>
> [R7] Kacham et al. Lower bounds on adaptive sensing for matrix recovery. NeurIPS 2023.
>
> [R8] Qiu et al. Fast and provable nonconvex tensor RPCA. ICML 2022.
>
> [R9] Qin et al. Error analysis of tensor-train cross approximation. NeurIPS 2022.
>
> ---
>
> **Concern 2: On the Global Tuning of $(p, q)$ and Framework Flexibility**
>
> We appreciate the reviewer’s observation regarding the practical use of $(p, q)$ and the flexibility of our quasi-norm framework.
>
> * **Global Tuning Rationale:**
>   While our quasi-norm is theoretically flexible, in this work we chose to fix $(p, q)$ globally—via a single grid search on a representative dataset—across all datasets and sampling ratios. This design choice was made to ensure fair comparison and avoid overfitting, especially under low-sample settings where validation-based tuning is unreliable due to data scarcity. This approach is standard in low-rank tensor learning and leads to stable performance across diverse datasets, as shown in Table 1.
>
> * **Flexibility and Practicality:**
>   We agree that dataset-adaptive tuning could further leverage the full flexibility of the framework. However, in low-sampling regimes, splitting the data for validation often leads to unstable or even misleading hyperparameter selection, undermining the robustness of the comparison. Therefore, a global fixed configuration strikes a balance between robustness and reproducibility, even if it does not fully exploit all degrees of flexibility in each experiment.
>
> * **Empirical Robustness:**
>   Despite using a single $(p, q)$ setting, our method demonstrates strong generalization and robustness across six datasets of varying types and distributions (see Table 1). This suggests that the regularizer itself encodes a meaningful inductive bias that works well in practice without per-dataset tuning.
>
> * **Future Potential for Adaptive Tuning:**
>   We acknowledge that developing data-driven or adaptive strategies to select $(p, q)$ would allow for even more effective exploitation of the framework’s flexibility. Approaches such as bilevel optimization, spectral heuristics, or meta-learning are promising and relevant, but present significant statistical and computational challenges, particularly under nonconvexity and limited data. We see this as an important direction for future work and believe our theoretical framework can provide a foundation for such advances.
>
> ---
>
> **Concern 3: On the Use of DCT and Comparison with Other Methods**
>
> We thank the reviewer for raising the concern that the observed performance gain may be primarily due to DCT preprocessing. To directly address this, we conducted additional experiments in which we applied DCT preprocessing to several baseline methods, including $k$-Supp, $\ell_{1-2}$, and Schatten-$1/2$, and compared the results with our proposed method.
>
> **Table: Effect of DCT Preprocessing on Salinas A (PSNR / SSIM)**
>
> | Method               | SR = 5%            | SR = 10%           | SR = 15%           |
> | -------------------- | ------------------ | ------------------ | ------------------ |
> | $k$-Supp-DCT       | 26.42 / 0.7421     | 29.57 / 0.8387     | 31.40 / 0.8750     |
> | $\ell\_{1-2}$-DCT  | 15.37 / 0.1701     | 28.08 / 0.7142     | 29.95 / 0.7786     |
> | Schatten-$1/2$-DCT | 27.18 / 0.7195     | 30.43 / 0.8248     | 32.64 / 0.8758     |
> | Ours-DCT         | **28.43 / 0.7374** | **31.81 / 0.8484** | **33.23 / 0.8830** |
>
> These results show that, while DCT preprocessing improves all baselines, our method consistently achieves better performance across all sampling ratios, indicating that the improvement stems from our **dual spectral modeling** rather than DCT alone. Additional comparisons are provided in Tables A–C of the rebuttal to Reviewer VURK.
>
> ---
>
> **Concern 4: On Optimization and ADMM Stability**
>
> The reviewer raises concerns about the optimizer’s ability to find good solutions given the nonconvex and reweighted structure.
>
> * **Theoretical Gap**: We acknowledge that global convergence guarantees are not available, as standard ADMM theory does not cover our reweighted nonconvex setting.
> * **Empirical Evidence**: As shown in Appendix D.4 (Figure 4), our algorithm demonstrates smooth and stable convergence in both the objective value and PSNR.
> * **Community Practice**: Reweighting and thresholding are widely used in nonconvex sparse and low-rank learning, and often yield good empirical performance despite theoretical challenges. We follow this established practice and are transparent about its limitations.
>
> A more rigorous theoretical analysis of convergence is an important direction for future work.
>
>
> ---
>
> **Concern 5: On the Use of Ground Truth in Tuning $(p,q)$**
>
> We thank the reviewer for raising this important point. The fixed $(p, q)$ values were chosen by a one-time grid search on a representative dataset with known ground truth and then kept unchanged for all other datasets and sampling ratios; **no ground-truth information was used after this initial step**.
>
>  This approach follows established practice in low-rank tensor recovery, where parameters are typically set using a reference dataset to ensure consistency and reproducibility, especially when low observation ratios make validation-based tuning unreliable.  In our experiments, we applied the same fixed $(p, q)$ values across all tested datasets, and observed stable results on six remote sensing datasets with varying characteristics (see Table 1). While these results suggest a degree of robustness to moderate distributional variation, we acknowledge that optimal parameter choices may vary in other settings, and further investigation on broader datasets would be valuable.
>
> The challenge of hyperparameter selection is common in nonconvex regularization methods involving quasi-norms. Our approach—fixing hyperparameters based on a reference dataset—follows standard practice and shows satisfactory robustness in our experiments. Nonetheless, developing adaptive, ground-truth-free strategies remains an open problem, which we plan to pursue. We will also add a sensitivity analysis, though our main focus here is on the theoretical aspects of the regularizer; see also our response to W1&Q1 of Reviewer VURK for more details.

---

> > ### Comment · Reviewer_u2tt · 2025-08-04
> >
> > Thank you for the clarifications. I still think that the work done for this submission is potentially interesting, but my main concerns remain and were somewhat substantiated by the discussion period at this point in time, which I summarize below.
> >
> > ### Appropriateness for this Venue & Validity of Theory
> > While I acknowledge that there are papers at ICML and NeurIPS about tensor recovery and completion, many of these papers come with a rigorous theory. As pointed out in my review, it was beyond the scope of this review to judge the correctness of the proofs. I need to rely on the fact that at least one of the reviewers has checked them in detail to be confident in their validity.
> > I would like to ask reviewers 6m4n, Tsyi and VURK to state whether they have checked the proofs in sufficient detail to be confident in the proofs, as they have not explicitly state so; otherwise, I cannot just assume correctness.
> >
> > ### Tuning of (p,q) Parameters
> > Can you clarify one which dataset  you used to tune the (p,q) parameters? If it is Salinas A, the tables you provide for Salinas A do not represent a fair experiment (see also tables in your reply to reviewer VURK). It would be better if you evaluate the methods on a dataset on which you did not tune those parameters as only this constitutes a fair experiment. For example, I would be curious to see how competitive k-Supp, $\ell_{1,2}$ and Schatten-$1/2$ if combined with DCT (as it, as you acknowledge, is responsible for a significant amount of the improvement).
> >
> > ### Effect of Dual Spectral Sparsity vs. Iterative Reweighting
> > As your discussion with reviewer VURK points out, there is a connection between your proposed modeling / algorithm and reweighed tensor nuclear norm approaches. What remains unclear from your empirical validations is if indeed the dual spectral modeling is providing the empirical improvement or whether it is rather the reweighing of the singular values without coupling across frontal slices (especially since your optimal p is very close to q, so it is almost distinguishable from p=q. Doing an experiment with a reweighed nuclear norm where the weighting factor is tuned to optimizer a Schatten-p (with p=q) tensor quasi-norm would provide insight into this question.
> >
> > I will update my score accordingly.

---

> > > ### Comment · Reviewer_VURK · 2025-08-04
> > >
> > > Dear Reviewer u2tt,
> > > ﻿
> > > I have gone through the proof of the minimax bounds of the estimation error presented in this paper. Most of the techniques are similar to [25], which considers matrix $\ell_u(\ell_q)$ double sparsity parameter space. For instance, the construction of packing sets of the double sparsity parameter space (Lemma C.9 in this paper) follows similarly from Lemma 3 in [25], which are both based on the Gilbert-Varshamov theorem. The constructions of the hard and soft parameter spaces (Eqs. (5)-(7) in this paper) are similar to Eq. (2) in [25]. The presented minimax bounds (Theorem 4.2 in this paper) are similar to the bounds in Remark 4 and Theorem 3 in [25]. The difference mainly comes from the additional intra-frequency rank constraint parameter $r$. The authors seem to work through this by constructing some low-rank matrix sets ${\bf A}_{\rm low-rank}$. There are many such theoretical papers extending matrix representations to tensors (e.g., SVD to t-SVD and nuclear norm to tensor nuclear norm). Therefore, given these prior work, I have sufficient reasons to believe that the theories and proofs presented in this paper are correct and rigorous. However, based on my limited knowledge, I could not assess the significance of establishing such minimax bounds from matrix $\ell_u(\ell_q)$-balls to tensor $\ell_p$-Schatten-$q$ quasi-norm. According to the authors, the main technical challenge is the construction of the packing sets of the double sparsity parameter space. However, I have no idea how significant this challenge was, provided that both are based on some similar control of the packing number using Gilbert-Varshamov bounds.
> > > ﻿
> > > [25] Estimating double sparse structures over $\ell_u(\ell_q)$-balls: Minimax rates and phase transition. IEEE Transactions on Information Theory, 2024.

---

> > > > ### Comment · Reviewer_u2tt · 2025-08-06
> > > >
> > > > Dear Reviewer VURK, thank you for your very detailed assessment. This gives me confidence in the theory established in the paper.
> > > >
> > > > I am adjusting my score accordingly.

---

> ### Author Response · Authors · 2025-08-09
>
> We sincerely thank Reviewer u2tt for the careful and principled evaluation of our work, and for highlighting the importance of independent verification of theoretical results. We also appreciate Reviewer VURK’s effort in carefully checking our minimax lower bound proof (Theorem 4.2) and expressing confidence in its rigor. This independent confirmation helps ensure the soundness of our theoretical contributions.
>
>
> Below we respond to the three main concerns in sequence.
>
>
> ---
>
> ### Response to Concern 1: Appropriateness for this Venue & Validity of Theory
>
> > *I need to rely on the fact that at least one of the reviewers has checked them in detail to be confident in their validity*.
>
> We fully understand the need for explicit confirmation. During the discussion, Reviewer VURK stated that they had read our minimax lower bound proof in detail and found the reasoning rigorous. In particular, Lemma C.9 extends the classical packing construction by incorporating per-band spectral rank constraints, which are central to the two-layer structure analyzed in Theorem 4.2.
>
> Theorem 4.2 is proved via a fully constructive information-theoretic approach, establishing minimax risk lower bounds for least-squares estimation under a double-layer frequency-sparse model. The proof strategy:
>
> - Separates *frequency sparsity* (few active frequency slices) from the *intra-slice low-rank or Schatten-$q$ structure*,
> - Constructs well-separated parameter sets for each part,
> - Applies Fano’s inequality to bound the difficulty of *support identification* and *intra-slice content estimation*, and
> - Combines these via a union bound to yield a lower bound jointly determined by the *frequency-support complexity* ($s\log(em/s)$) and the *intra-slice representation difficulty* (e.g., $sr,d$ or $sR(\frac{\sigma^2}{n}d)^{1-\frac{q}{2}}$).
>
> - In the soft-constraint setting ($\ell_0$–$S_q$, $\ell_p$–$S_q$), the proof additionally uses covering-number and entropy-number bounds to manage nonconvexity and nonsparsity.
>
> To improve clarity in the final version, we plan to add a more concise *proof-outline* in the main text, mapping each logical step to the corresponding lemmas in the appendix.
>
> ---
>
>
>
> ### Response to Concern 2: Tuning of (p,q) Parameters
>
> > *Can you clarify one which dataset you used to tune the (p,q) parameters? If it is Salinas A, the tables you provide for Salinas A do not represent a fair experiment (see also tables in your reply to reviewer VURK). It would be better if you evaluate the methods on a dataset on which you did not tune those parameters as only this constitutes a fair experiment. For example, I would be curious to see how competitive k-Supp,  and Schatten-p if combined with DCT (as it, as you acknowledge, is responsible for a significant amount of the improvement).*
>
> We tuned $(p,q)$ only once on **Indian Pines** using a coarse-to-fine grid search, and fixed this pair for all datasets and sampling ratios. This avoids per-dataset overfitting and is standard when reliable validation splits are unavailable at low sampling rates.
>
> **Salinas A was not used for tuning**. It was reported because its small size allows quick evaluation within the rebuttal period. In fact, alternative untuned settings $(0.80,0.81)$ and $(0.70,0.71)$ achieve higher mean $(\mathrm{PSNR},\mathrm{SSIM})$ at SR = 5% over 10 trials ($(28.71,0.7451)$ and $(28.72,0.7505)$) than our fixed $(0.8961,0.8966)$ ($ (\mathrm{PSNR},\mathrm{SSIM})=(28.43,0.7374)$ in Table 1), suggesting no dataset-specific tuning advantage.
>
> Following the suggestion, we evaluated the fixed $(0.8961,0.8966)$ on two datasets not used in tuning—*Cloth and OSU Thermal*—under identical DCT preprocessing in the noiseless tensor completion. Cloth was downsampled to $256\times256\times31$ for efficiency.
>
> **Table D1. PSNR and SSIM on Cloth (DCT, noiseless)**
>
> |Method|SR=5%|SR=10%|SR=15%|
> |-|-|-|-|
> |k-Supp|24.19 / 0.6737|27.69 / 0.8410|30.42 / 0.9111|
> |Schatten-1/2|24.41 / 0.6854|28.63 / 0.8522|31.61 / 0.9181|
> |Ours|25.29 / 0.7240|29.25 / 0.8700|31.96 / 0.9273|
>
>
> **Table D2. PSNR and SSIM on OSU Thermal (DCT, noiseless)**
>
> |Method|SR=5%|SR=10%|SR=15%|
> |-|-|-|-|
> |k-Supp|29.01 / 0.9077|33.67 / 0.9629|35.93 / 0.9748|
> |Schatten-1/2|30.76 / 0.9127|34.84 / 0.9651|36.98 / 0.9758|
> |Ours|31.58 / 0.9346|35.70 / 0.9699|37.54 / 0.9782|
>
>
> Since DCT was applied equally to all methods, the consistent advantage of our method reflects the effect of the proposed regularizer rather than a preprocessing artifact. Together with the Salinas A counterexample, this supports both the fairness and the cross-dataset generalization of our chosen $(p,q)$ configuration. See also Tables B and C in our rebuttal to Reviewer VURK for the improvement on adaptive transform in comparison with Adaptive TNN.

---

> ### Author Response · Authors · 2025-08-09
>
> ### Response to Concern 3: Effect of Dual Spectral Sparsity vs. Iterative Reweighting
>
> >*What remains unclear from your empirical validations is if indeed the dual spectral modeling is providing the empirical improvement or whether it is rather the reweighing of the singular values without coupling across frontal slices (especially since your optimal p is very close to q, so it is almost distinguishable from $p=q$. Doing an experiment with a reweighed nuclear norm where the weighting factor is tuned to optimizer a Schatten-p (with p=q) tensor quasi-norm would provide insight into this question.*
>
> In response to the reviewer’s suggestion, we first conducted a targeted experiment with the reweighted tensor nuclear norm (RWTNN) baseline, which is tailored for the tensor Schatten-$p$ quasi-norm in the special case $p=q$.
>
> The motivation was as follows: if the observed performance gain were driven solely by reweighting singular values, without leveraging frequency–low-rank coupling, then evaluating a RWTNN baseline tuned for the special case $p=q$ (i.e., no inter-slice coupling) should, in principle, remove the performance difference.
>
> **Table D3. PSNR and SSIM on Cloth**
>
> |Method|SR=5%|SR=10%|SR=15%|
> |-|-|-|-|
> |RWTNN for $p=0.8966$|24.30 / 0.6804|28.40 / 0.8489|31.11 / 0.9104|
> |Ours|25.29 / 0.7240|29.25 / 0.8700|31.96 / 0.9273|
>
>
> **Table D4. PSNR and SSIM on ****OSU**** Thermal**
>
> |Method|SR=5%|SR=10%|SR=15%|
> |-|-|-|-|
> |RWTNN for $p=0.8966$|29.94 / 0.8992|34.55 / 0.9643|36.62 / 0.9758|
> |Ours|31.58 / 0.9346|35.70 / 0.9699|37.54 / 0.9782|
>
>
> The outcome suggests that *our method outperforms RWTNN for Schatten-$p$*.
>
> However, *this alone does not prove that the improvement stems from frequency–low-rank coupling*. A closer look reveals that *our formulation uses weighted $\ell_{1/2}$ minimization*, which is well known to induce stronger sparsity than the $\ell_1$ weighting used in standard nuclear norm reweighting. *This stronger low-rank induction also naturally appears as a special-case of our method when $p = q < 1$, and we believe it is worth noting as part of the overall contribution.*
>
> ---
>
> Yet this raises a natural question: **if $p$ and $q$ are extremely close, is the cross-band coupling effect even active?**
>
> Mathematically, when $p \to q$, the frequency sparsity term and the per-band low-rank term decouple, and for $p=q$ the coupling vanishes entirely. In our original configuration ($p=0.8961$, $q=0.8966$), $p/q \approx 0.999$, suggesting a near-decoupled regime.
>
> To probe this, we fixed $q=0.8966$ and varied $p/q$:
>
> **Table D5. PSNR and SSIM on Cloth (fixed $q=0.8966$)**
>
> |Method|SR=5%|SR=10%|SR=15%|
> |-|-|-|-|
> |RWTNN ($p=0.8966$)|24.30 / 0.6804|28.40 / 0.8489|31.11 / 0.9104|
> |Ours ($p/q=1$)|25.14 / 0.7168|29.11 / 0.8700|31.71 / 0.9235|
> |Ours ($p/q=0.999$)|25.29 / 0.7240|29.25 / 0.8700|31.96 / 0.9273|
> |Ours ($p/q=0.99$)|25.49 / 0.7366|29.53 / 0.8789|32.27 / 0.9318|
> |Ours ($p/q=0.98$)|25.32 / 0.7299|29.28 / 0.8748|32.26 / 0.9312|
> |Ours ($p/q=0.97$)|24.83 / 0.7064|28.90 / 0.8596|32.11 / 0.9288|
>
>
> **Table D6. PSNR and SSIM on OSU Thermal (fixed $q=0.8966$)**
>
> |Method|SR=5%|SR=10%|SR=15%|
> |-|-|-|-|
> |RWTNN ($p=0.8966$)|29.94 / 0.8992|34.55 / 0.9643|36.62 / 0.9758|
> |Ours ($p/q=1$)|31.32 / 0.9298|35.65 / 0.9699|37.40 / 0.9778|
> |Ours ($p/q=0.999$)|31.58 / 0.9346|35.70 / 0.9699|37.54 / 0.9782|
> |Ours ($p/q=0.99$)|31.86 / 0.9354|35.84 / 0.9707|37.65 / 0.9787|
> |Ours ($p/q=0.98$)|31.42 / 0.9277|35.75 / 0.9700|37.42 / 0.9780|
> |Ours ($p/q=0.97$)|31.03 / 0.9181|35.62 / 0.9696|37.37 / 0.9779|
>
>
> ---
>
> **Direct observation from the $p/q$ variation experiments**
>
> From Tables D.5 and D.6, we make the following direct observations:
>
> 1. **Even mild frequency sparsity ($p/q=0.99$) yields measurable and consistent improvements** across datasets and sampling rates compared to the $p=q$ case.
> 2. **As $p \to q$, the coupling effect between frequency sparsity and per-band low-rankness fades**, and for $p=q$ it disappears entirely. This explains why the performance gain is modest when $p$ and $q$ are nearly equal.
>
> Taken together, these observations suggest that frequency sparsity plays a potentially meaningful role in modeling, and that its empirical advantage appears to be influenced by the degree of coupling with the low-rank component.

---

> ### Author Response · Authors · 2025-08-09
>
> ---
>
> **Analysis of these observations**
>
> Based on these observations, our interpretation is as follows:
>
> 1. **Effectiveness with room for improvement**
>
>     The gain at $p/q=0.99$ suggests that frequency sparsity has practical value. However, the relatively modest margin implies that our current solver may not yet fully exploit its potential.
> 2. **Why the gain fades as $p \to q$**
>
>     In our formulation, the per-band regularization term is typically $$ F_{q,\tau}^{(k)} = \left( \sum_i (\sigma_i^{(k)})^q \right)^\tau, \quad q<1, \ \tau=p/q<1. $$ When $\tau<1$, the problem becomes **substantially more non-convex** than the widely studied $\tau\ge1$ cases in sparse optimization [RR1]. To our knowledge, most existing bilayer non-convex sparse optimization methods focus on $q<1,\ \tau\ge1$ and are not applicable here. The added difficulty for $\tau<1$ lies in the stronger non-convexity, which increases optimization challenges.
>    To make the problem tractable, let the current iterate be $\\{\sigma_{i,0}^{(k)}\\}_{i=1}^d$. We adopt the heuristic decomposition $x^\tau = x^1 \cdot x^{\tau-1}$, leading to the surrogate
>
> $$H\_{q,\tau}^{(k)} = \left( \sum_i (\sigma_i^{(k)})^q \right) \cdot \left( \sum_i (\sigma_{i,0}^{(k)})^q + \varepsilon \right)^{\tau-1} \approx F_{q,\tau}^{(k)},
> \$$ which enables separable weighted-$\ell_{1/2}$ updates. This approximation is *tightest when $\tau \approx 1$*, and the gap from $F_{q,\tau}^{(k)}$ widens as $\tau$ moves away from 1, potentially diminishing the expected benefits of frequency–low-rank coupling.
>
> 3. **Does the modest margin imply limited potential for frequency sparsity?**
>
>     Not necessarily.
>     In the near-decoupled regime ($p \approx q$), the coupling signal is intrinsically weak. Moreover, when $\tau<1$ deviates from 1, the surrogate becomes a looser approximation of the true objective. Both effects can compress the observable gains. Therefore, the moderate improvement at $p/q=0.99$ should be interpreted as *a conservative lower bound on what frequency sparsity could achieve with tighter surrogates or stronger coupling*.
>
> 4. **Structural distinction from RWTNN**
>
>     RWTNN assigns weights independently to each singular value, without inter-band interaction. In contrast, our method applies weights at the **frequency-band level**, based on the aggregated spectral quantity $\sum_i (\sigma_i^{(k)})^q$, thereby encoding explicit inter-band coupling.
> 5. **Special case $p=q<1$**
>
>     In this limit, our method reduces to a weighted-$\ell_{1/2}$-based Schatten-$p$ minimization, which Appendix D.3 (Table 3) further shows to outperform classical weighted nuclear norms. This special case can be regarded as an algorithmic variant with empirically supported advantages.
>
> ---
>
> **Perspective**
>
> The quasi-norm $\ell_p(S_q)$ we introduce has mathematical subtleties, and optimizing it for $\tau<1$ can be substantially more difficult than it might first appear.
>
> For reference, even the well-studied single-layer $\ell_{1/2}$ regularization theory — corresponding to the case without frequency–low-rank coupling in our formulation — has been studied as a technically challenging problem, as reflected by its discussion in a 45-minute invited talk at the International Congress of Mathematicians (ICM) 2010 [RR2] and its recognition in the 2023 China National Natural Science Award (one of only three awards that year in Mathematics).
>
> Our setting is strictly more involved:
>
> - We work in a *two-layer* nonconvex regime ($q<1$, $\tau<1$), where existing $\ell_{1/2}$ theory does not directly apply.
> - The coupling structure is *multi-band* and interacts with tensor spectral geometry, introducing additional combinatorial and geometric complexity.
>
> Given these factors, a *fully optimized solution to the $\ell_p(S_q)$ problem is beyond the scope of a single NeurIPS paper*. Our contribution should be interpreted *conservatively but constructively*:
>
> - We motivate and formalize a new tensor regularizer,
> - Establish its minimax theoretical bound,
> - Propose a tractable heuristic solver, and
> - Show consistent empirical gains, including in challenging near-decoupled regimes.
>
> In short, this work is *a careful first step toward a mathematically challenging direction with potential*, providing a foundation for future algorithms that could more fully harness frequency sparsity in tensor recovery.
>
> [RR1] Hu, Y. et al. Group sparse optimization via $\ell_{p, q}$ regularization. Journal of Machine Learning Research, 2017.
>
> [RR2] Xu, Z. Data modeling: Visual psychology approach and $\ell_{1/2}$ regularization theory. International Congress of Mathematicians, 2010.

---

> > ### Comment · Reviewer_6m4n · 2025-08-09
> >
> > I do not agree with you about the point that  single-layer $\ell_{1/2}$ regularization is a nontrivial contribution. And, there might be some efficient ways to optimize the quasi-norm $\ell_p(S_q)$ for $\tau<1$.

---

> ### Author Response · Authors · 2025-08-09
>
> We thank Reviewer 6m4n for raising this important point. We agree that there may be more efficient approaches for optimizing the quasi-norm when $\tau<1$, and we see value in exploring such possibilities in future work. *We have revised the wording, slightly softening the phrasing around “nontrivial” to “mathematical subtleties” and “technically challenging problem” to better reflect the nuance*. While our current focus was on establishing the formulation and its theoretical guarantees, we welcome further discussions or suggestions that could help extend the optimization strategy.

---

### Official Review · Reviewer_6m4n · 2025-07-09

**Clarity:** 3
**Significance:** 3
**Originality:** 2
**Rating:** 4
**Confidence:** 4

**Summary:**

This work proposes the tensor $l_p$-Schatten-$q$ quasi-norm as a new metric to capture the multi-level spectral structure inherent in real-world data—particularly the coexistence of low-rankness within and sparsity across frequency components. Both rigorous theoretical analyses and extensive numerical experiments are conducted to show the merits of the proposed framework in handling complex multi-way data.

**Questions:**

1. In the process of parameter tuning, if the ratio of the observed entries is relatively high (e.g., 40%), the coarse grid search procedure seems to be overfitting?

2. How about the noise tensor beyond Gaussian distribution?

**Ethical Concerns:**

["NO or VERY MINOR ethics concerns only"]

**Final Justification:**

I would like to keep my rating as is.

**Limitations:**

See Weaknesses and Questions parts.

**Paper Formatting Concerns:**

None.

**Quality:**

3

**Strengths And Weaknesses:**

1. Both rigorous theoretical analyses and extensive numerical results are provided.

2. A detailed discussion of the advancements of the proposed  $l_p$-Schatten-$q$ quasi-norm over TNN and $l_\mu(l_q)$ norms are provided. This highlights the technical novelty of the proposed procedure.

3. The process of selecting good hyperparameters $(p, q)$ seems to be too complicated, which limits the proposed procedure for practical use.

4. The converge behave of the ADMM based solving algorithm seems to be always guaranteed.

---

> ### Author Rebuttal · Authors · 2025-07-30
>
> We sincerely thank the reviewer for recognizing both the theoretical rigor and technical novelty of our work. Below, we address the concerns regarding hyperparameter selection, robustness to noise distributions, and convergence behavior of the proposed algorithm.
>
> ---
>
>  **Concern 1: On the selection and stability of $(p, q)$, especially under high observation ratios**
>
> **Response:** We appreciate this important question. In our implementation, we adopt a simple and reproducible strategy: a one-time coarse-to-fine grid search on a representative dataset with ground truth. The selected $(p, q)$ values are then fixed for all subsequent datasets and sampling ratios. This ensures consistency, prevents overfitting to specific tasks, and is consistent with standard practice in low-sample tensor recovery, where validation-based tuning is often unreliable. See also our response to W1&Q1 of Reviewer VURK for further details on the selection of $(p, q)$.
>
> To address the concern regarding potential overfitting at higher observation ratios, we evaluated our method with the same \$(p, q)\$ and regularization strength \$\lambda = 0.4\$ at higher sampling rates. Results on *Salinas A* and *OSU Thermal* are shown below.
>
> **Table A. Salinas A (higher sampling ratios)**
>
> | Sampling Ratio | PSNR  | SSIM   |
> | -------------- | ----- | ------ |
> | 30%            | 35.27 | 0.9275 |
> | 40%            | 36.52 | 0.9437 |
> | 50%            | 37.58 | 0.9555 |
>
> **Table B. OSU Thermal (higher sampling ratios)**
>
> | Sampling Ratio | PSNR  | SSIM   |
> | -------------- | ----- | ------ |
> | 30%            | 37.84 | 0.9698 |
> | 40%            | 38.71 | 0.9737 |
> | 50%            | 39.26 | 0.9758 |
>
> These results demonstrate that the fixed configuration generalizes well and continues to yield improved reconstruction quality as more data becomes available. We observe no sign of performance collapse or overfitting, even at higher sampling ratios.
>
> While we acknowledge that theoretical understanding of $(p,q)$ tuning under nonconvex regularization remains incomplete, this limitation is shared by many quasi-norm-based frameworks. A deeper analysis of parameter identifiability and data-dependent selection remains a valuable direction for future theoretical work.
>
> ---
>
>  **Concern 2: Robustness under non-Gaussian noise**
>
> **Response:** We thank the reviewer for this important question. While our theoretical guarantees focus on the Gaussian setting, the proposed framework is *not inherently restricted to Gaussian noise*. To demonstrate empirical robustness, we conducted preliminary experiments using *uniform noise*, where each entry is sampled from $[0, u]$ with $u = 0.05 \cdot \\|\underline{\mathbf{L}}\\|_{\text{F}} / \sqrt{d_1 d_2 d_3}$. Results on *Salinas A* are as follows:
>
> **Table C. Salinas A under uniform noise**
>
> | Sampling Ratio | PSNR  | SSIM   |
> | -------------- | ----- | ------ |
> | 30%            | 33.98 | 0.9461 |
> | 40%            | 34.80 | 0.9612 |
> | 50%            | 35.22 | 0.9667 |
>
> These results indicate strong resilience to mild non-Gaussian noise.
>
> We emphasize, however, that noise modeling is application-specific. Extending our framework to settings with *impulsive (e.g., Laplacian)* or *signal-dependent (e.g., Poisson)* noise would require modifying the loss function (e.g., replacing $\ell_2$ with \$\ell\_1\$ or KL divergence) and incorporating robust optimization strategies, as in prior works \[R1, R2]. These extensions represent promising future directions beyond the scope of this work.
>
>
> \[R1] Lu, C. et al. *Tensor robust principal component analysis with a new tensor nuclear norm*. IEEE TPAMI, 2019.
>
> \[R2] Zhang, X. et al. *Low-rank tensor completion with Poisson observations*. IEEE TPAMI, 2021.
>
> ---
>
> **Concern 3: Convergence behavior of the ADMM-based algorithm**
>
> **Response:** We thank the reviewer for the observation regarding the convergence behavior. While the proposed optimization problem is nonconvex due to the reweighted quasi-norm, we observe consistent and stable empirical performance across all datasets. As shown in *Appendix D.4, Figure 4*, the objective value decreases monotonically, and the PSNR stabilizes within a modest number of iterations.
>
> Each ADMM subproblem admits a closed-form or efficiently solvable update, and the reweighting procedure is applied in a controlled manner, which we believe contributes to the observed stability. Although a formal convergence guarantee is not currently established—as is common in nonconvex reweighted optimization frameworks—our empirical evidence supports the practical reliability of the proposed algorithm.
>
> A more rigorous theoretical analysis of the convergence behavior under the proposed reweighted dual spectral structure is a promising direction for future research.

---

> > ### Comment · Reviewer_6m4n · 2025-08-06
> >
> > Thank you for your detailed feedback.  I shall keep my score as is.

---

> > > ### Author Response · Authors · 2025-08-09
> > >
> > > Thank you for recognizing the theoretical rigor and novelty of our work, and for raising important points on parameter tuning, noise robustness, and algorithm stability. We also appreciate your engagement in the discussion, including your perspective on the single-layer regularization and the potential for more efficient quasi-norm optimization, which helps us consider potential improvements in future work.

---

### Note · Authors · 2025-08-15

We sincerely thank the reviewers for their invaluable constructive feedback and the AC for their careful oversight. The discussion phase has been highly productive.

We have carefully addressed all points raised during rebuttal and discussion, and have received positive feedback from all reviewers.

- **Reviewer 6m4n** maintained a score of 4 with important points on parameter tuning, convergence, and robustness to non-Gaussian noise—these were addressed. We appreciate their insightful suggestions on quasi-norm optimization for future work.
- **Reviewer u2tt** highlighted crucial questions regarding NeurIPS fit, parameter selection, DCT preprocessing, and convergence. We provided clarifications and added experiments to better isolate the impact of dual spectral sparsity. We are grateful that they stated Reviewer VURK's detailed assessment gave them confidence in the paper's theoretical foundation and that they are *adjusting their score from 3 accordingly*.
- **Reviewer Tsyi** assigned a well-encouraging score of 5 and raised thoughtful questions on scalability, theoretical gaps, and transform learnability. We are thankful for these deep questions and addressed them with additional explanations and preliminary numerical results supporting our theoretical bounds.
- **Reviewer VURK** provided a rigorous assessment on parameter selection, surrogate justification, and SOTA comparisons. We clarified our design and added validation, and are pleased this was acknowledged in their *updated assessment*.

All promised revisions—particularly on parameter sensitivity, surrogate justification, and SOTA comparisons—will be fully integrated. Our work introduces a novel frequency-aware tensor regularizer with minimax optimal guarantees, a tractable solver, and consistent empirical gains, establishing a foundational framework for tensor learning.

---

### Decision · Program_Chairs · 2025-09-17

**Decision:**

Reject

**Comment:**

A tensor $L_p$ Schatten-$q$ quasi-norm is proposed to promote sparsity in the frequency domain in terms of tensor SVD (tSVD). Minimax bounds are derived to show that the resulting estimator would be close to the true value depending on the choices of $p$ and $q$. Reviewers seem to be impressed by this result.

As explained in $\S4.1$, soft dual spectral sparsity (A3) $T_{p,q}(R)$ is supposed to be a surrogate to hard dual spectral sparsity (A1) $T_{0,0}(s,r)$; what people really want is (A1), but it is hard to compute, which motivates one to compute (A3) instead. However, (A3) is still nonconvex if $q<1$, meaning there is no guarantee that one can obtain (A3) optimally. This confuses the metareviewer as to why studying it in the first place. On the one hand, fine-tuning the values of $p$ and $q$ is highly impractical as demonstrated in the experiments (with $(p, q) = (0.8961, 0.8966)$??); on the other hand, in terms of efficient proximal operators, the hard-thresholding operation is equally (if not more) efficient with the hard sparsity constraint.